# Regulating the scaling relationship for high catalytic kinetics and selectivity of the oxygen reduction reaction

Wanlin Zhou[1], Hui Su [1,2] ✉, Weiren Cheng [1,3], Yuanli Li[4], Jingjing Jiang[1], Meihuan Liu[1], Feifan Yu[5], Wei Wang[5], Shiqiang Wei [1] & Qinghua Liu [1] ✉

The electrochemical oxygen reduction reaction (ORR) is at the heart of modern sustainable energy technologies. However, the linear scaling relationship of this multistep reaction now becomes the bottleneck for accelerating kinetics. Herein, we propose a strategy of using intermetallic-distance-regulated atomic-scale bimetal assembly (ABA) that can catalyse direct O–O radical breakage without the formation of redundant *OOH intermediates, which could regulate the inherent linear scaling relationship and cause the ORR on ABA to follow a fast-kinetic dual-sites mechanism. Using in situ synchrotron spectroscopy, we directly observe that a self-adjustable N-bridged $Pt = N_2 = Fe$ assembly promotes the generation of a key intermediate state (Pt–O–O–Fe) during the ORR process, resulting in high reaction kinetics and selectivity. The well-designed $Pt = N_2 = Fe$ ABA catalyst achieves a nearly two orders of magnitude enhanced kinetic current density at the half-wave potential of 0.95 V relative to commercial Pt/C and an almost 99% efficiency of 4-electron pathway selectivity, making it one of the potential ORR catalysts for application to the energy device of zinc–air cells. This study provides a helpful design principle for developing and optimizing other efficient ORR electrocatalysts.

The oxygen reduction reaction (ORR) is the core of renewable energy conversion technology and plays an irreplaceable role in emerging electrochemical energy devices such as metal–air batteries and fuel cells[1-3]. However, the sluggish kinetics of the ORR that occurs on the cathode involves multiple-step proton-coupled electron transfer, which has a significant impact on improving the overall efficiency of energy conversion devices[4-7]. Catalysts based on the noble metal platinum (Pt, US $36,084 kg$^{-1}$) with metal loadings up to $400\ \mu g_{Pt}\ cm^{-2}$ are the most practical for boosting the energy output efficiency of ORR-driven energy devices, but their application is still hindered by the scarcity of noble metal resources and limited stability under desirable

high voltages (>0.6 V)[2,8]. Alloying Pt with a secondary metal can reduces the usage of Pt while improving the performance[1,9,10]. However, the amount of Pt is still not less than 40 at.%, and the secondary metal leaches away gradually under ORR conditions, resulting in rapid performance losses, which sufficiently limits large-scale commercialization[11,12]. Therefore, the design of low-loading active noble-metal ORR electrocatalysts with fast reaction kinetics and long-term durability is both desirable and essential for efficient energy conversion and storage.

According to the widely accepted conventional single-site mechanism of multiple-step proton-coupled electron transfer, the

[1]National Synchrotron Radiation Laboratory, University of Science and Technology of China, Hefei 230029 Anhui, P. R. China. [2]School of Materials Science and Engineering, Anhui University, Hefei 230601 Anhui, P. R. China. [3]Institute for Catalysis, Hokkaido University, Sapporo 001-0021, Japan. [4]Fundamental Science on Nuclear Wastes and Environmental Safety Laboratory, Southwest University of Science and Technology, Mianyang 621010 Sichuan, P. R. China. [5]School of Chemistry and Chemical Engineering, Key Laboratory for Green Processing of Chemical Engineering of Xinjiang Bingtuan, Shihezi University, Shihezi 832003, China. ✉e-mail: suhui@ustc.edu.cn; qhliu@ustc.edu.cn

ORR process involves multiple reaction intermediates, including $^*O_2$, $^*OOH$, and $^*O$[13,14]. Of note, the binding energies of these intermediates over the single sites are closely related and follow the scaling relationship[15,16]. This means that the binding energy of each intermediate cannot be independently adjusted because of the insurmountable scaling relation. Consequently, the oxygen intermediates ($^*O_2$ and $^*OOH$) with large binding energy directly lead to sluggish kinetics and require a high overpotential to drive the reaction[17,18]. As a result, the scaling relationship in the ORR now becomes the bottleneck for further improving its performance. To overcome this dilemma, an alternative route is to bypass the formation of oxygen intermediates ($^*OOH$) and promote direct O–O radical breakage without extra reaction intermediates ($^*OOH$) by a dual-site mechanism (M–O–O–M)[19,20]. Under this circumstance, the direct breakage of O–O on the dual-sites alleviates the complex dynamics of multiple reaction intermediates. More importantly, the dual-site mechanism is a fast 4-electron reaction path, since the sluggish kinetics of $^*OOH$ production is overcome. Therefore, it can be anticipated that to further improve the performance and selectivity of current ORR electrocatalysis, the dual-site mechanism with a fast four-electron reaction path is an extremely ideal choice. However, the realization of this dual-sites mechanism requires stringent requirements on the geometric and electronic configuration of the active site. Intermetallic distances of too great a length cause the single-site mechanism to be accompanied by M–OOH, which is common in single-site catalysis, and distances that are too short, such as in metal nanoparticles, easily trigger the emergence of two-electron reaction paths[21-23]. Thus, an appropriate interatomic spacing in two adjacent metallic active sites is mandatory for the dissociation of $^*O_2$ and the triggering of O–O radical breakage (M–O–O–M) to demonstrably suppress M–OOH production and limiting the two-electron reaction path selectivity during the ORR process.

Herein, to satisfy the dual-site mechanism design rule for high kinetics and selectivity of four-electron ORR, we develop a N-bridged $Pt=N_2=Fe$ atomic-scale bimetal assembly (ABA) model catalyst with tailored geometry configuration by a controllable "amino functionalized carbon nanoflakes" strategy. The amino functionalization of carbon nanoflakes with controllable size can anchor Fe atoms adjacent to Pt atoms with suitable intermetallic atomic spacing for the dual-sites mechanism. At the atomic level, in situ synchrotron radiation X-ray absorption fine structure (XAFS) spectroscopy reveals that the self-adjustable $Pt=N_2=Fe$ bimetal assembly promotes the evolution of $O_2$ molecules to O–O radicals, which then adsorb to N-bridged metal sites in a dual-binding mode ($M_1$–O–O–$M_2$). The sensitive in situ synchrotron radiation Fourier transform infrared spectroscopy (SR-FTIR) monitors the formation of key intermediate states (Pt–O–O–Fe) in the $Pt=N_2=$ Fe ABA catalysts under operating conditions, demonstrating that the ORR follows a dual-sites mechanism without the production of $^*OOH$ intermediates. The well-designed $Pt=N_2=Fe$ ABA catalyst exhibits a prominent catalytic kinetics with the kinetic current density ($J_k$) of 5.83 mA cm$^{-2}$ at the half-wave potential of 0.95 V, which is nearly two orders of magnitude higher than that of Pt/C. Moreover, the efficiency of 4-electron pathway selectivity for the $Pt=N_2=Fe$ ABA catalyst approaches ~99%, delivering good performance in zinc–air cells with a peak power density of 198.4 mW cm$^{-2}$ at 315.2 mA cm$^{-2}$. Thus, the $Pt=N_2=Fe$ ABA catalyst becomes a promising alternative to the commercial Pt/C-based air cathode (172.1 mW cm$^{-2}$ at 253 mA cm$^{-2}$).

## Results and discussion

### Synthesis and characterization of atomic-scale bimetal assembly

The nitrogen-rich catalyst precursor was synthesized via a unique two-step strategy of functionalized carbon nanoflakes (CNF–NH$_2$) and a subsequent pyrolysis process. The preparation scheme can be seen in Supplementary Fig. 1. The pyrene ($C_{16}H_{10}$) as feedstock was nitrated into trinitropyrene with a high activation state in hot $HNO_3$ and then

the amino-functionalized carbon nanoflakes (CNF–NH$_2$) can be obtained after the nitro group (-NO$_2$) was almost completely replaced by amino group (-NH$_2$) in amino functionalization step (Supplementary Fig. 2). For more heteronuclear metal dual-sites to be synthesized, the precursor solution containing $Fe^{3+}$ and $[PtCl_6]^{2-}$ was prepared in glycol solvent to promote the effective affinity between the two hetero-electric metal groups[24,25]. The pretreated bimetallic cations chelated stably with the functionalized amine groups (–NH$_2$) during the freeze-drying process and uniformly dispersed on the surface of CNFs–NH$_2$. Finally, the atomic-scale bimetal assembly (ABA) catalyst was obtained by a pyrolysis process in an inert atmosphere. The Fourier transform infrared spectroscopy (FTIR) characterization results (Supplementary Fig. 3) reveal that the metal was successfully modified on the amino group of CNF-NH$_2$[26,27]. And after annealing at 700 °C, the metal remains the strong interaction with nitrogenous matrix (Supplementary Fig. 4)[28]. The atomically monometallic sites catalysts (Pt AMS and Fe AMS) are synthesized for reference in similar process, except that the precursor solution is replaced with $H_2PtCl_6$ or $FeCl_3 \cdot 6H_2O$ aqueous solution, respectively.

The layered morphology of the as-synthesized $Pt=N_2=Fe$ ABA catalyst is illustrated by the electron micrograph in Fig. 1a and Supplementary Fig. 5. The atomically dispersed metal sites in $Pt=N_2=Fe$ ABA catalyst is revealed by Cs-corrected scanning transmission electron microscopy high-angle annular dark field (STEM-HAADF) images (Fig. 1b and Supplementary Fig. 6). Based on the fact that the intensity of atoms in the STEM-HAADF image highly depends on their atomic number ($Z$), intensity analysis was performed at four dual-site regions in STEM-HAADF images to clarify the composition of the dual-sites structure (Fig. 1c and Supplementary Fig. 6d, e)[29]. All the two bright spots along the arrow show different intensity, confirming that the dual-sites is composed of Pt and Fe atoms. The statistical analysis results show that the atomic distance between Pt and Fe is about 2.83–2.91 Å. And the statistical analysis based on 500 randomly selected metal sites reveals that the metal dual-sites structure dominates about 72.6% (Supplementary Fig. 7). In the X-ray diffraction (XRD) pattern (Fig. 1d), all the samples show only a dominant diffraction peak (~25°) of carbon, and no crystalline metal nanoparticles are observed. And the uniform distribution of Pt, Fe, N and C elements in the $Pt=N_2=Fe$ ABA catalyst can be reflected by the energy-dispersive X-ray (EDX) mapping images (Fig. 1e, Supplementary Fig. 8)

The surface chemical states were detected by X-ray photoelectron spectroscopy (XPS). The XPS survey spectrum reveals that the $Pt=N_2=Fe$ ABA catalyst contains Pt, Fe, N, C and O elements (Supplementary Fig. 9), and the metal contents of Pt and Fe in $Pt=N_2=Fe$ ABA determined by XPS are 2.93 and 1.34 wt%, respectively, which are close to the inductively coupled plasma–optical emission spectrometry (ICP–OES) results (inset of Supplementary Fig. 9). The O 1$s$ XPS spectrum in Supplementary Fig. 10b shows the absence of M-O bond, suggesting the inexistence of metal oxide species or O-coordinated monoatomic metals in $Pt=N_2=Fe$ ABA sample[30,31]. The N 1$s$ XPS spectrum was fitted into a typical peak at 399.35 eV, which could be attributed to the N atom interaction with metal sites (Pt/Fe–N$_x$) in $Pt=N_2=Fe$ ABA (Supplementary Fig. 10a and Supplementary Table 1)[32]. In addition, the fitting results of Pt 4$f$ and Fe 2$p$ XPS spectra for $Pt=N_2=Fe$ ABA show that the metal atoms have a positive charge state with no zero-valence species appearing (Supplementary Fig. 11, Supplementary Table 2–3). The oxidation states of Pt and Fe were further analyzed quantitatively based on the X-ray absorption near-edge structure (XANES) spectra (Supplementary Fig. 12a)[33]. The normalized Pt $L_3$-edge XANES spectra by subtracting the spectra of $Pt=N_2=Fe$ ABA and $PtO_2$ sample to that of Pt foil reference are shown in Supplementary Fig. 12b. The valence state of Pt in the $Pt=N_2=Fe$ ABA sample can be calculated as +3.36 (Supplementary Fig. 12c) through the area integral of normalized white-line peak, which is consistent with the valence state analysis in XPS characterization[34]. The valence

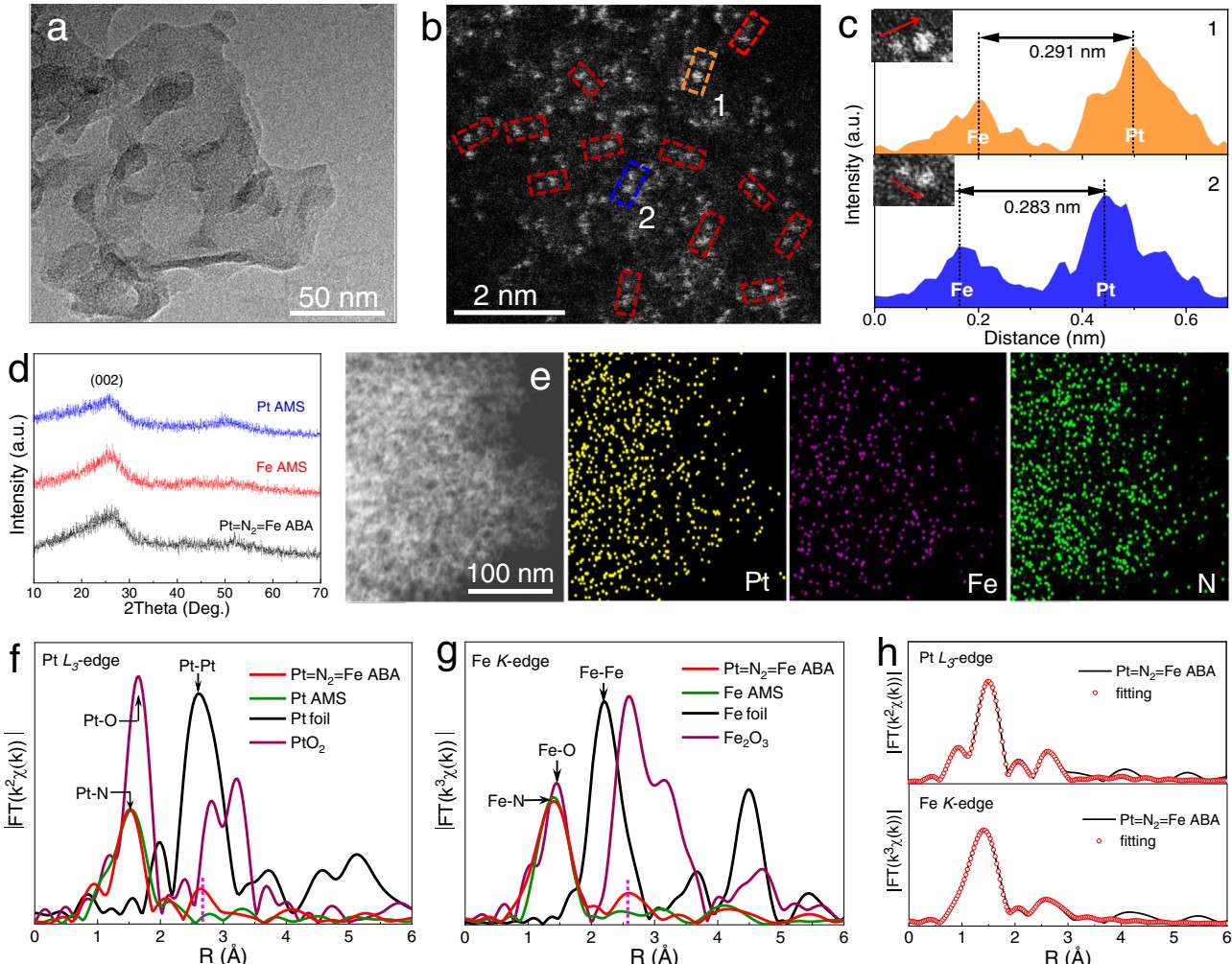

**Fig. 1 | Morphology and structural characterizations. a, b** TEM (**a**) and HAADF-STEM (**b**) images of the Pt = N₂ = Fe ABA catalyst. **c** Intensity profile along Line 1 and Line 2 in **b**. **d** XRD patterns of Pt=N₂=Fe ABA, Fe AMS and Pt AMS. **e** TEM-EDS mapping images for Pt = N₂ = Fe ABA. **f–h** FT-EXAFS spectra of Pt $L_3$-edge (**f**) and Fe $K$-edge (**g**) for Pt = N₂ = Fe ABA catalyst and reference samples, and the corresponding fitting curves for Pt = N₂ = Fe ABA (**h**).

state of Fe can be quantitatively evaluated as +2.40 based on the difference value ($\Delta E_0$) of the absorption edge, which validates the XPS results (Supplementary Fig. 13)[35].

The extended XAFS (EXAFS) was detected to further identify the coordination environment of metal sites in Pt = N₂ = Fe ABA. For the Fourier transformed (FT) EXAFS spectra of the Pt $L_3$-edge, the Pt = N₂ = Fe ABA shows a dominant peak at 1.55 Å (Fig. 1f), similar to that in Pt AMS samples, which is typically assigned to the first shell of Pt–N coordination. Concurrently, in contrast with the reference Pt AMS sample, another characteristic peak is clearly observed at 2.63 Å, suggesting the presence of a metal coordination of the second shell. It is noteworthy that the peak position of metal coordination is slightly longer than that of the Pt–Pt shell in Pt foil and shorter than the second shell of Pt–O–Pt in PtO₂. The Fe $K$-edge FT-EXAFS spectrum of Pt = N₂ = Fe ABA demonstrates a predominant peak at 1.42 Å (Fig. 1g), which is similar to that in Fe AMS samples, confirming the presence of Fe–N coordination. Moreover, the Pt = N₂ = Fe ABA catalyst presents a characteristic peak at 2.60 Å in the second shell that is obviously longer than the Fe–Fe bond in Fe foil, which may be attributed to the second shell of Fe–N–Pt coordination. Combined with the FT-EXAFS results of the Pt $L_3$-edge, it can be demonstrated that the Pt and Fe atoms in the catalyst are assembled with an appropriate intermetallic distance rather than bonding directly. Thus, we infer that the two nearby Pt and Fe atoms may coordinate with bridged N atoms at the second shell. To obtain the quantitative coordination information in the Pt = N₂ = Fe ABA catalyst, FT-EXAFS curve-fitting analysis was performed based on the structures of dual-sites (72.6%) and monometallic sites (25.3%) (Fig. 1h, Supplementary Fig. 14 and Supplementary Table 4–5). The fitting results of both Pt $L_3$-edge and Fe $K$-edge FT-EXAFS spectra show that the bond length ($R$) of the metal coordination is around 2.86 Å, corresponding to the atomic spacing analysis results in Cs-corrected STEM-HAADF images. Furthermore, in the best fitting results of Pt $L_3$-edge EXAFS, the coordination numbers (CNs) for the Pt–N bond and the second shell of Pt–N–Fe are 4.1 and 1.1, respectively (Supplementary Table 4). The best fitting result for Fe $K$-edge EXAFS shows CNs of 4.2 and 1.0 for the Fe–N bond and Fe–N–Pt coordination, respectively (Supplementary Table 5). The above results imply that each of the Pt and Fe atoms coordinates with four N atoms to form an M–N₄ configuration, and the adjacent Pt and Fe atoms are connected by two co-shared bridge N atoms to form a Pt₁Fe₁–N₆ geometric configuration.

## Electrochemical oxygen reduction performance

The electrocatalytic ORR activities of the as-synthesized Pt = N₂ = Fe ABA, Pt AMS, Fe AMS catalysts and the commercial Pt/C were evaluated on a rotating disk electrode (RDE) system. The cyclic voltammetry curves were recorded for the synthesized platinum-based catalysts (Pt = N₂ = Fe ABA and Pt AMS) in N₂-saturated 0.1 M KOH electrolyte, which present no hydrogen adsorption and/or desorption

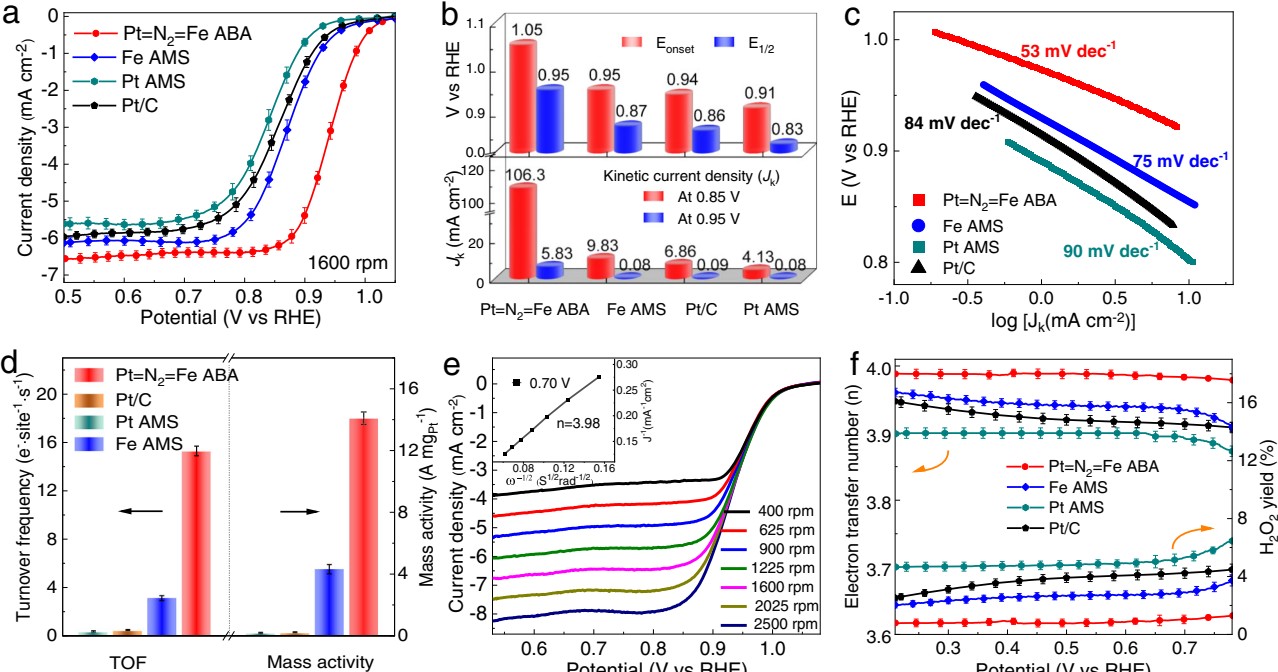

**Fig. 2 | Electrochemical oxygen reduction performance measurements. a** ORR polarization plots of Pt = N$_2$ = Fe ABA, Pt/C, Pt AMS and Fe AMS. **b** The performance parameters of half-wave potential ($E_{1/2}$), onset potential ($E_{onset}$) and kinetic current density ($J_k$). **c** Tafel plots, **d** turnover frequency (TOF) and mass activity for Pt = N$_2$ = Fe ABA and reference catalysts. **e** Polarization plots at regular rotation rates for Pt = N$_2$ = Fe ABA with the inset showing the derived Levich plots. **f** ORR pathway selectivity parameters for the catalysts involved. The error bars were determined by the deviation of three individual measurements.

characteristic due to the monoatomic dispersion of platinum in the synthesized catalysts (Supplementary Fig. 15)[36]. The linear sweep voltammetry (LSV) curves illustrated in Fig. 2a obviously show that Pt = N$_2$ = Fe ABA has a superior performance compared with the reference catalyst. As quantified in Fig. 3b, Pt = N$_2$ = Fe ABA possesses a half-wave potential ($E_{1/2}$) of 0.95 V vs. RHE and onset potential ($E_{onset}$) of 1.05 V vs. RHE, which are 90 and 100 mV higher than those of commercial Pt/C (0.86 and 0.95 V), also outperforming those of Pt AMS, Fe AMS and almost all reported ORR catalysts (Supplementary Table 6). As for the reaction kinetics, the Pt = N$_2$ = Fe ABA presents an ultrahigh kinetic current density ($J_k$) of 5.83 mA cm$^{-2}$ at 0.95 V (Fig. 2b), nearly two orders of magnitude higher than that of commercial Pt/C ($J_k$, 0.09 mA cm$^{-2}$). The accelerated current density of Pt = N$_2$ = Fe ABA at 0.85 V is still dozens of times higher than that of commercial Pt/C (Pt = N$_2$ = Fe ABA, 106.3 mA cm$^{-2}$; Pt/C, 9.83 mA cm$^{-2}$). The superior ORR kinetics of Pt = N$_2$ = Fe ABA is further confirmed by the smaller Tafel slope (53 mV dec$^{-1}$), which is significantly smaller than those of commercial Pt/C (84 mV dec$^{-1}$), Fe AMS (75 mV dec$^{-1}$) and Pt AMS (90 mV dec$^{-1}$) (Fig. 2c). Based on the roughly equal numbers of active sites in the as-synthesized catalysts, the improved catalytic performance of the Pt = N$_2$ = Fe ABA is mainly attributed to the formed Pt–N-Fe dual-sites. Notably, the criteria of turnover frequency (TOF) and mass activity (MA), which can characterize the intrinsic activity of the catalyst, were calculated and shown in Fig. 2d. Pt = N$_2$ = Fe ABA displays a TOF of 15.3 e$^-$·site$^{-1}$·s$^{-1}$ and a large MA of 14.1 A mg$_{Pt}^{-1}$ at 0.90 V vs. RHE, 32 and 61 times higher than that of commercial Pt/C reference (0.48 e$^-$·site$^{-1}$·s$^{-1}$, 0.23 A mg$_{Pt}^{-1}$), and the high MA of platinum in Pt = N$_2$ = Fe ABA exceeds most of the reported platinum-based catalyst (Supplementary Table 7). The cyclic voltammetry curves of Pt = N$_2$ = Fe ABA catalyst and the reference samples with the scanning rates from 10 to 50 mV s$^{-1}$ are shown in Supplementary Fig. 16. As a result, the Pt = N$_2$ = Fe ABA catalyst has the highest electrochemically accessible surface area (ECSA) of 210.5 m$^2$ g$^{-1}$, corresponding to the highest ORR activity[37].

Catalytic selectivity is an important index used to evaluate the performance of electrocatalysts, especially for electrocatalytic oxygen reduction. The ORR pathway was assessed via polarization curve measurements at different rotation rates (Fig. 2e)[38]. The electron transfer number ($n$) was calculated to be ~3.98 according to the Levich equation applied in the diffusion-controlled region (inset of Fig. 2e and Supplementary Fig. 17), proving that the Pt = N$_2$ = Fe ABA catalyst favours the 4e$^-$ pathway. Moreover, the measurement conducted on the rotating ring-disk electrode (RRDE) also shows that n for Pt = N$_2$ = Fe ABA is ~3.98 (Fig. 2f), which is better than Pt/C and reference samples, confirming that the side reaction of the 2e$^-$ pathway was effectively suppressed. Importantly, the H$_2$O$_2$ yield of Pt = N$_2$ = Fe ABA was measured to be below 1.5%, which represents a high selectivity for the 4-electron ORR[39]. The rapid O−O bond breaking are realized over Pt = N$_2$ = Fe ABA for an efficient 4-electron ORR process. In addition, stability during operation is particularly important for the commercial application of catalysts. The Pt = N$_2$ = Fe ABA catalyst demonstrates reliable stability with better carbon oxidation and methanol resistance in chronoamperometry measurements, in which the $E_{1/2}$ only drops 15 mV and the capacitance increases <4% after continuous operation for 100 h (Supplementary Fig. 18–20). The Pt = N$_2$ = Fe ABA catalyst also exhibits advanced performance in acidic electrolyte in comparison with the monometallic sites catalysts (Pt AMS and Fe AMS) and the standard commercial Pt/C (Supplementary Fig. 21), which suggests the synergistic effect of the dual-sites in Pt = N$_2$ = Fe ABA for catalyzing the ORR process. These results demonstrate that the Pt = N$_2$ = Fe ABA catalyst can serve as a promising electrocatalyst with prominent electroreduction activity and selectivity.

**In situ XAFS analysis**
The in situ XANES and EXAFS spectra, which can reflect the dynamic evolution of active sites during the actual reaction process, were employed to explain the nature of the high ORR activity[40,41]. Pt $L_3$-edge XANES in Fig. 3a and the corresponding local magnification of Fig. 3b show that the white-line intensity increases obviously when a potential

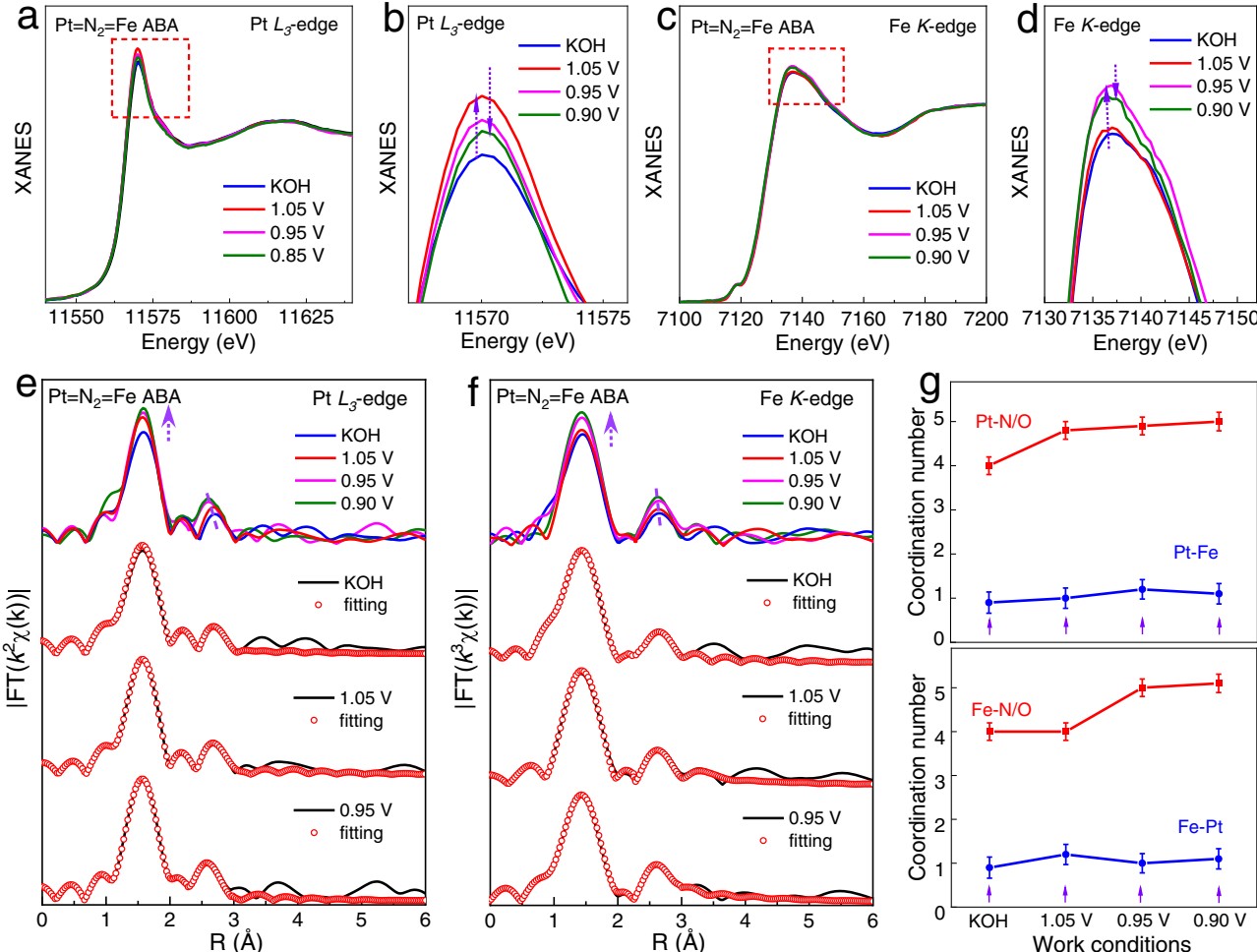

**Fig. 3 | The structure evolution identified by in situ XAFS. a–d** The Pt $L_3$-edge (**a**) and Fe $K$-edge (**c**) XANES spectra, and **b**, **d** are the corresponding local magnifications of the white-line peaks. **e**, **f** The Pt $L_3$-edge (**e**) and Fe $K$-edge (**f**) FT-EXAFS spectra and the corresponding fitting curves under different potentials. **g** The fitting results of the coordination number.

of 1.05 V is applied, arising from the higher oxidation state caused by the emergence of Pt sites coordinated with oxygen species. With the application of oxygen reduction potentials, the intensity of the white line decreases, implying an electron transition from nearby atoms to Pt sites for an accelerated catalytic reaction process. Interestingly, for the Fe $K$-edge XANES (Fig. 3b, c), the white-line intensity changes similarly with potential hysteresis, indicating the coordination with oxygen at the Fe sites are closely adjacent to the Pt sites. This implies that Pt and Fe are synergistic active centres that act as adsorption/desorption sites for reactive species.

To further clarify the structural evolution of the bimetal assembly active sites, FT-EXAFS spectra are presented for both Pt $L_3$-edge and Fe $K$-edge. For the FT-EXAFS spectra of Pt $L_3$-edge under different reaction conditions in Fig. 3e, the dominant peak at ~1.55 Å, which can be attributed to the first-shell of Pt–N/O coordination, obviously increases in intensity, especially from at KOH to at 1.05 V. Analogously, in the FT-EXAFS spectra of Fe $K$-edges (Fig. 3f), we observe the same phenomenon that the intensity of the dominant peak at ~1.42 Å increases with a voltage of 0.95 V, indicating that both Pt and Fe metal sites are involved in the reaction and have a similar coordination evolution in the ORR process. To quantify the evolution of the local structure at Pt = N₂ = Fe dual-sites, the corresponding FT-EXAFS curve fitting was conducted, as shown in Fig. 3e–g, Supplementary Figs. 22–25, and Supplementary Tables 4–5. The best fitting results of the Pt $L_3$-edge show that the coordination numbers (CNs) of the Pt–N path is 4.0, which is consistent

with the Pt–N₄ configuration obtained from the ex-situ Pt = N₂ = Fe ABA sample. When a potential of 1.05 V is applied, considering the inevitable adsorption of oxygen species during the reaction, one additional Pt–O coordination (2.04 Å) is added. It cannot be ignored that Fe–O coordination is added with a CN of 1.2 at 0.95 V, based on the fitting of Fe $K$-edge FT-EXAFS curves as Fe–N₄ coordination under KOH conditions. Comprehensive analysis of the fitting results for the dual-sites shows that the oxygen molecules can be adsorbed separately on the appropriately distanced Pt and Fe atoms, likely promoting the intermetallic assembling-sites co-coupling of dioxygen intermediates (–O–O–) in a dual-site adsorption form (Pt–O–O–Fe). It is of particular interest that the intermetallic distance of the second-shell Pt–N–Fe coordination at ~2.6 Å in both the Pt $L_3$-edge and Fe $K$-edge FT-EXAFS curves tends to shrink under operating conditions, indicating that the N-bridged Pt = N₂ = Fe assembly can be self-adjustable during the reaction process. This behaviour facilitates the formation of Pt–O–O–Fe active intermediate state at the Pt = N₂ = Fe dual-sites. Two adjacent metal atoms in Pt = N₂ = Fe assembly each binding an O atom of a dioxygen species (–O–O–) can provide a strong driving force for direct O–O bond breakage, regulating the scaling relationship of multiple reaction intermediates for a fast 4-electron ORR process.

## In situ SR-FTIR analysis

In situ SR-FTIR, which can sensitively capture the reactive species and identify the reaction mechanism, was performed under ORR

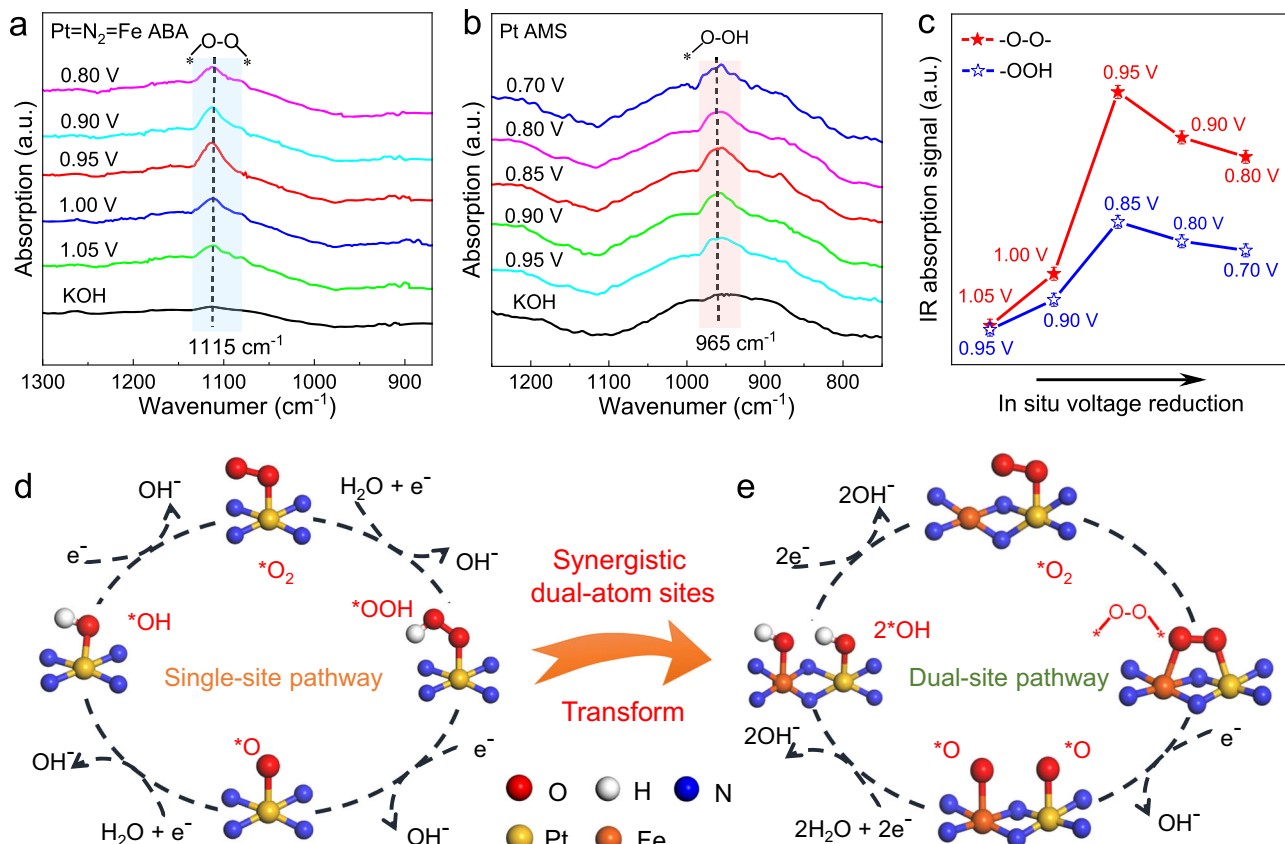

**Fig. 4 | The reaction pathway identified by in situ SR-FTIR. a, b** 800–1300 cm$^{-1}$ range of in situ SR-FTIR characterizations for Pt = N$_2$ = Fe ABA (**a**) and Pt AMS (**b**). **c** FTIR absorption stretching at 1115 cm$^{-1}$ for Pt = N$_2$ = Fe ABA and at 965 cm$^{-1}$ for Pt AMS. **d, e** Schematic of the ORR mechanism for Pt = N$_2$ = Fe ABA (**e**) and Pt AMS (**d**).

operating conditions[42–44]. As shown in Fig. 4a, an IR absorption band located at 1115 cm$^{-1}$ is observed with a gradually applied potential. It is known that the absorption bands within the range of 1060–1150 cm$^{-1}$ are associated with the IR absorption of O–O[45,46]. Consequently, this voltage-dependent vibration suggests that the key intermediate O–O is generated over the Pt = N$_2$ = Fe dual-sites to form a Pt–O–O–Fe configuration during the ORR process. Combined with the in situ XAFS results, the dual-sites of Pt and Fe favouring the adsorption of O$_2$ molecules directly evolve into O–O radicals. As a reference, Fig. 4b shows the FTIR measurement results of the Pt AMS sample under the same conditions. A new IR absorption band appears around 965 cm$^{-1}$ after the applied potential exceeds 0.95 V vs. RHE. Considering that the infrared vibration peaks of the *OOH species adsorbed on single-metal sites typically lie at -900 cm$^{-1}$, the absorption vibration at 965 cm$^{-1}$ can be assigned to the emergence of *OOH intermediate species[47,48]. Meanwhile, the in situ SR-FTIR measurement was also conducted for the Fe AMS sample to make a well comparison (Supplementary Fig. 26). The newly formed infrared vibration peak located at 980 cm$^{-1}$, which is similar to the Pt AMS sample, informing the formation of *OOH on the single iron site within Fe AMS. It can be seen that *O–O* radical with obviously different peak position (1115 cm$^{-1}$) only appears for Pt = N$_2$ = Fe ABA, suggesting that O$_2$ is effectively activated into *O$_2$ and cleaves directly. The synergistic effect of dual-sites with suitable intermetallic spacing in Pt = N$_2$ = Fe ABA can greatly reduce the reaction barrier by promoting the generation of O–O radicals and further improve the kinetic activity of the ORR. To clarify the real reaction mechanism of Pt = N$_2$ = Fe ABA under operating conditions, the vibration peak intensities of *O–O* (1115 cm$^{-1}$) and *OOH (965 cm$^{-1}$) relative to the applied potential are statistically shown in Fig. 4c. The vibration intensity of the –O–O– band increases rapidly as the

ORR potential exceeds 1.05 V. In contrast, the intensity of *OOH for the Pt AMS catalyst only shows a slow enhancement after 0.95 V. This phenomenon confirms that the Pt = N$_2$ = Fe dual-sites within the atomic-scale bimetal assembly catalyst significantly accelerate the evolution of O$_2$ into –O–O– radicals to form the Pt–O–O–Fe intermediate structure, enabling the ORR to occur at a lower overpotential. Meanwhile, the Pt–O–O–Fe active phase can effectively cleave the O–O bond without the production of *OOH to regulate the inherent scaling relationship for fast kinetics.

The traditional single-sites and innovative dual-sites reaction pathway mechanisms between the active sites and key intermediates are summarized in Fig. 4d, e. For the traditional single-sited reaction mechanism, the intermediates of *OOH are adsorbed on the single active sites by an end-on bonding state, which inevitably hinders the reaction kinetics by the difficult formation and cleavage of *OOH intermediates[32,49]. Delightfully, on the bimetal assembly dual-sites within the Pt = N$_2$ = Fe ABA catalyst, O$_2$ molecules are adsorbed on the Pt sites to form *O$_2$ intermediates first. Then, under the driving voltage, the distance between the adjacent Pt and Fe atoms is contracted, and –O–O– radicals are formed to form the highly active Pt–O–O–Fe structure. The strong stretching effect of bimetal assembly sites on –O–O– radicals promotes O–O bond cleavage to form the M–O* meso-phase. Compared with the single-sites reaction for Pt AMS and Fe AMS, this dual-sites reaction bypasses the generation of the *OOH intermediate, which significantly lowers the energy barrier by regulating the scaling relationship for the binding energies of multiple intermediates. Moreover, by virtue of the intermetallic bimetal assembly sites, the O–O bond is accelerated to cleavage, which essentially reduces the energy barrier of the 4-electron reaction process and achieves high activity and selectivity of catalytic ORR.

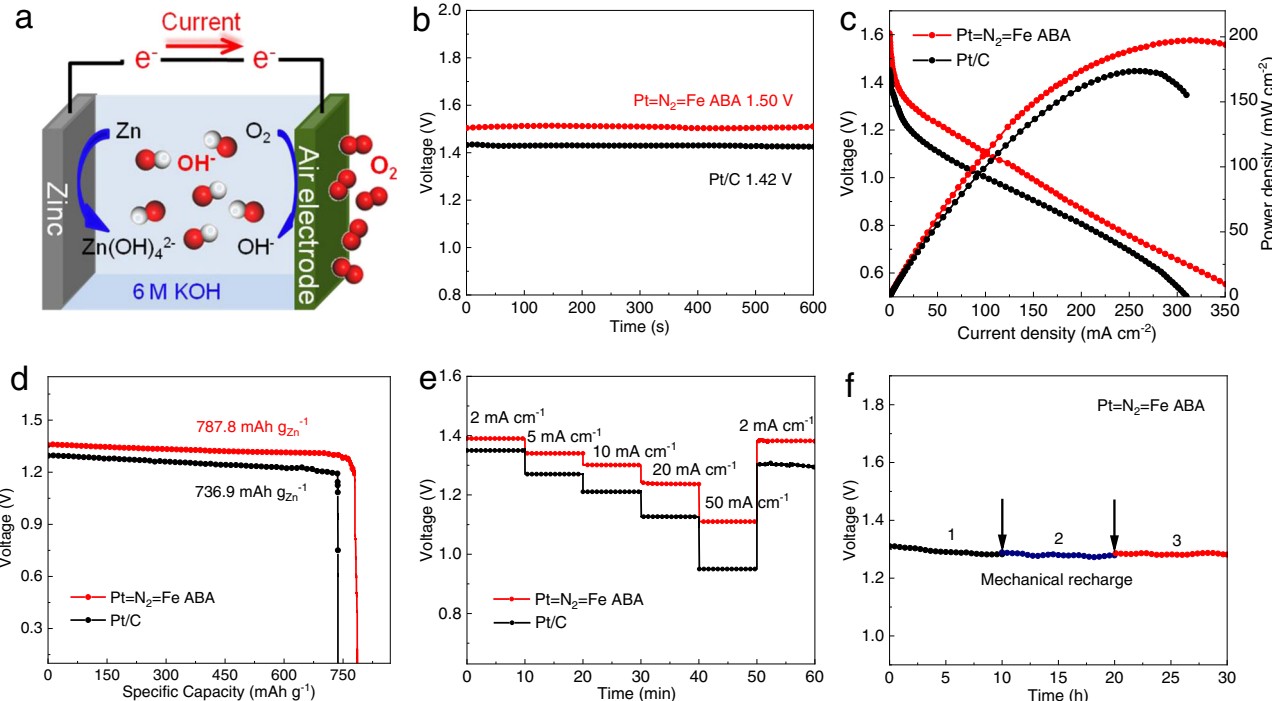

**Fig. 5 | Zn–air battery (ZAB) performance. a** Device diagram of the Zn–air battery. **b** Open-circuit voltage (OCV) of the ZAB based on Pt = N₂ = Fe ABA and the reference Pt/C catalyst. **c** Polarization curves and power density of the assembled ZAB. **d** Specific capacities of the assembled ZAB. **e** Galvanostatic discharge curves under different current densities. **f** Long-term discharge curve with three recharging cycles.

## Applications in renewable energy devices

We assembled the Pt = N₂ = Fe ABA as the cathode of a primary Zn–air battery (ZAB) to evaluate the efficiency of the catalyst in renewable energy devices (Fig. 5a). As shown in Fig. 5b, the Pt = N₂ = Fe ABA-incorporated Zn–air cell exhibits an open-circuit voltage (OCV) of 1.50 V, which is higher than that of the counterpart Pt/C-based Zn–air cell (1.42 V). Notably, a maximum power density of 198.4 mW cm⁻² at 315.2 mA cm⁻² is achieved for the ZAB incorporated with the Pt = N₂ = Fe ABA catalyst due to the favourable mass transfer (Fig. 5c), outperforming the Pt/C-based ZAB (172.1 mW cm⁻² at 253 mA cm⁻²) and ZABs made from other catalysts (Supplementary Table 8). Furthermore, the ZAB with the Pt = N₂ = Fe ABA air cathode delivers a specific capacity of 787.8 mAh g$_{Zn}^{-1}$ at a discharge current density of 10 mA cm⁻², accounting for 96.1% of the theoretical capacity of ZAB (~820 mAh g$_{Zn}^{-1}$) (Fig. 5d). The enhanced performance proves that the Pt = N₂ = Fe ABA catalyst can deliver efficiency adequate for justifying incorporation into Zn–air cells. The galvanostatic discharge measurements under different current density platforms are shown in Fig. 5e. The cell voltage responses of the ZAB with Pt = N₂ = Fe ABA are obviously higher than those of the Pt/C cell at each current density, and the Pt = N₂ = Fe ABA cell voltage returns to its original level as the current density reverts to 2 mA cm⁻². This suggests the considerable rate performance and robust stability of Pt = N₂ = Fe ABA as a cathode in a ZAB. In addition, this Pt = N₂ = Fe ABA-based ZAB can work stably through mechanical recharging and requires only replenishment of the consumed zinc anode and electrolytes. No apparent decline in the output voltage is observed after three charging cycles (Fig. 5f). These results reveal that a unique Pt = N₂ = Fe ABA catalyst which maintains satisfactory catalytic activity in practical operation has great application prospect in industrial ORR.

In summary, a highly efficient and selective N-bridged Pt = N₂ = Fe bimetal assembly catalyst with appropriate intermetallic atomic spacing and unique electronic structure was developed by a controllable amino functionalized carbon nanoflakes strategy. In combination with in situ XAFS and SR-FTIR techniques, we identify that the key intermediate state (Pt–O–O–Fe) was successfully formed over self-adjustable N-bridged Pt = N₂ = Fe bimetal assembly sites under working conditions, which efficiently catalyses the ORR following a dual-sites mechanism, promoting the generation and fast cleavage of O–O radical intermediates without the formation of the traditional reactive product *OOH. The as-obtained Pt = N₂ = Fe ABA catalyst delivers a quite appreciable 4-electron selectivity of ~99% and a high kinetic current density which exceeds the commercial Pt/C by two orders of magnitude. Interestingly, the Pt = N₂ = Fe ABA catalyst guarantees high efficiency and robust stability in zinc-air batteries. Our results provide a useful design principle for developing highly active and selective ORR catalysts.

## Methods
### Synthesis of functionalized CNF–NH₂
The functionalized CNF–NH₂ was first synthesized as a precursor. Typically, 1 g pyrene (Aladdin, 99%) was dissolved in 80 ml nitric acid, refluxed and stirred at 80 °C for 12 h, cooled to room temperature, and centrifuged and washed in deionized water until neutral. The nitrated product 1,3,6-trinitropyrene was obtained by vacuum drying. 1,3,6-Trinitropyrene (300 mg) was dissolved in a mixed solution of 55 ml deionized water and 5 ml concentrated ammonia water (30%, Sigma) and ultrasonicated in cold water for 4 h. After the homogenized suspension was formed, it was poured into a Teflon-lined autoclave and heated at 200 °C for 10 h before cooling to room temperature. The solution was filtered through a 0.22 μm microporous membrane to remove the insoluble carbon products and then concentrated to ~20 mL by rotary evaporation to obtain the CNF–NH₂ concentrated solution.

### Pt = N₂ = Fe ABA synthesis
Typically, a mixed solution was made by adding 5.75 mg of H₂PtCl₆ and 2.27 mg of FeCl₃ to the glycol solvent. The mixture was ultrasonicated and mechanically agitated for 20 min to promote the effective association between electropositive Fe³⁺ and electronegative [PtCl₆]²⁻.

Then, 20 ml of concentrated CNF–NH$_2$ solution was added to the resulting mixed solution, sonicated in cold water for 15 min, quickly frozen with liquid nitrogen and dried. The product was mixed with urea at a mass ratio of 1:10, heated to 700 °C at a rate of 5 °C/min in a tubular furnace under a N$_2$ atmosphere, and held at that temperature for 2 h. Finally, the powder was stirred in 3 M HCl for 2 h, then thoroughly cleaned with deionized water and dried in a vacuum drying oven at 65 °C.

## Pt AMS and Fe AMS synthesis

The synthesis process for Pt atomically monometallic site (Pt AMS) catalyst and Fe atomically monometallic site (Fe AMS) catalyst is similar to that of the preparation of the Pt=N$_2$=Fe ABA electrocatalyst. The notable difference is that the separate aqueous solution of H$_2$PtCl$_6$ or FeCl$_3$ is mixed directly with the prefabricated ammonia-rich carrier (CNF–NH$_2$), and the metal sources are replaced with 11.5 mg of H$_2$PtCl$_6$ or 5.94 mg FeCl$_3$ for the synthesis of Pt AMS and Fe AMS, respectively.

## Characterization

JSM-6700F and JEM-2100F electron microscopes were employed to conduct SEM and TEM measurements at 5 kV and 200 kV, respectively. HAADF-TEM was conducted on a JEM-ARM200F instrument. XRD and XPS were performed on a Philips X'Pert Pro Super X-ray diffractometer with Cu Kα radiation and an ESCALAB MKII diffractometer with Mg Kα radiation, respectively.

## Electrochemical measurements

All electrochemical measurements for ORR performance were carried out on a CH Instruments CHI760E with a standard three-electrode cell. O$_2$-saturated 0.1 M KOH was used as the electrolyte, and graphite rods and Ag/AgCl were used as the counter electrode and reference electrode, respectively. The working electrode was an RDE with a disk diameter of 3 mm or an RRDE with a disk diameter of 5 mm. For the preparation of the catalyst ink for the RDE test, 2.5 mg catalyst powder was dispersed in a mixed solution of 750 μl DI water, 250 μl ethanol and 20 μl Nafion™ (5 wt%) and ultrasonicated for 30 min. The catalyst ink was pipetted onto the GC surface with a catalyst loading of ~0.1 mg cm$^{-2}$.

The mass activity is calculated by:

$$MA = J_k / M_{metal} \tag{1}$$

(the kinetic current densities $J_k$, mA cm$^{-2}$; the metal loading density $M_{Pt}$, μg$_{metal}$ cm$^{-2}$), the metal loading on electrode of 2.9 μg$_{Pt}$ cm$^{-2}$ in Pt=N$_2$=Fe ABA, 20 μg$_{Pt}$ cm$^{-2}$ in commercial Pt/C, 5.9 μg$_{Pt}$ cm$^{-2}$ in Pt AMS and 1.96 μg$_{Fe}$ cm$^{-2}$ in Fe AMS.

The turnover frequency (TOF) is calculated by the following formula:

$$TOF = \frac{J_k}{\frac{W_{metal}}{M_{metal}} \times N_A \times m_{catalystloading}} \tag{2}$$

Where $W_{metal}$ and $M_{metal}$ are the mass fraction (wt%) and molar mass (g mol$^{-1}$). $m_{loading}$ is the loading mass of the catalyst on RDE (g cm$^{-1}$). N$_A$ is Avogadro's number (6.02 × 10$^{23}$ mol$^{-1}$). The Pt=N$_2$=Fe pair was regarded as a unit of synergistic active site, and the dual-sites and single Pt/Fe active sites were taken into account in a statistical proportion (about 3:1).

For the tests of zinc-air batteries, custom-fabricated electrochemical cells with a two-electrode system were used. Pt=N$_2$=Fe ABA or Pt/C catalyst loaded onto Teflon-coated carbon fibre paper (1 cm$^2$) was used as the air cathode (catalyst loading was 1 mg cm$^{-2}$). Zn foil was used as the anode after polishing (1 cm$^2$). All the tests were carried out

in O$_2$-saturated 6.0 M KOH at room temperature with CHI 760E instruments.

## In situ SR-FTIR measurements

All the in situ SR-FTIR data were collected at BL01B of the National Synchrotron Radiation Laboratory (NSRL, China). An FTIR spectrometer, a KBr beam splitter, an MCT detector and an external microscope (Bruker Hyperion 3000) were used. The instrument adopts the reflection mode of vertical incident infrared light, the measurement width of the infrared spectrum is 600–4000 cm$^{-1}$, and the spectral resolution is 0.25 cm$^{-1}$. The custom-fabricated FTIR electrochemical cell was a three-electrode system, in which Ag/AgCl and platinum wire were used as the reference electrode and counter electrode, respectively. A 1 cm$^2$ carbon cloth loaded with catalyst was used as the working electrode and pressed onto the ZnSe crystal window with a micron gap. The electrolyte was 0.1 M KOH saturated with O$_2$ and circulated through a peristaltic pump.

## In situ XAFS measurements

All XAFS data were collected at BL11B station of Shanghai Synchrotron Radiation Facility (SSRF), China. The storage ring of SSRF operates at 3.5 GeV with a maximum current of 210 mA. The electrochemical in situ XAFS measurements were carried out in a custom-fabricated three-electrode system with a 1 × 1 cm$^2$ carbon cloth loaded with Pt=N$_2$=Fe ABA catalyst as the working electrode and 0.1 M continuously O$_2$-saturated KOH solution as the electrolyte. For the in situ XAFS spectral data acquisition of Pt $L_3$-edge (11564 eV) and Fe $K$-edge (7112 eV), we calibrated the positions of absorption edges ($E_0$) by using Pt foil and Fe foil standard samples, respectively, and all spectra were collected in the same beam time by fluorescence mode to ensure comparability.

## Strategies of XAFS curve Fittings

The ARTEMIS module implemented in IFEFFIT was used to perform the EXAFS quantitative curve fittings. The fittings for the Pt $L_3$-edge of Pt=N$_2$=Fe ABA sample were done for $k^2$–weighted $\chi(k)$ functions in a $k$-range of 2.5-11.5 Å$^{-1}$ and Fourier-transformed to real ($R$) space by Hanning windows at d$k$ = 1.0 Å$^{-1}$ within an $R$-range of 1.0–3.1 Å. The number of independent points is $N_{ipt} = 2\Delta k \times \Delta R/\pi = 2 \times (11.5 - 2.5) \times (3.1 - 1.0)/\pi = 12$. For the ex situ Pt=N$_2$=Fe ABA sample, the coordination peaks at around 1.55 Å assigned to the Pt–N coordination with N atoms that derived from the N-rich –NH$_2$ functionalized support. And based on the analysis of the dominant heteronuclear dimer atomic spacing in Cs-corrected STEM-HAADF images and the location of metal coordination peak, the peak at 2.65 Å in Fourier-transformed curves should be assigned to Pt–N–Fe coordination. Therefore, two separate Pt–N and Pt–N–Fe scattering paths were included for fitting. As for the in situ Pt=N$_2$=Fe ABA sample under 1.05, 0.95 and 0.90 V, the first coordination peaks show an obviously increased strength and location shift compared to the sample in KOH solution without potential applied condition. Considering that the prominent peak at ~1.6 Å can be assigned to the M–N/O coordination and the possible presence of oxygen-associated species, an additional Pt–O path was included during the fitting. The Debye–Waller factors ($\sigma^2$) and bond length ($R$) were treated as adjustable parameters for all comditions in all paths, the energy shift ($\Delta E_O$) for different paths under the same condition was set equal to reduce the number of adjustable fitting parameters, and the coordination numbers ($N$) of Pt–N under in situ conditions were fixed with reference to the ex situ sample. For the in situ samples under 1.05, 0.95 and 0.90 V, the number of adjustable parameters was $N_{para} = 9 < N_{ipt}$.

Similarly, the fittings for the Fe $K$-edge of Pt=N$_2$=Fe ABA sample were done in the $R$-space within the $R$-range of 1.0–3.1 Å for $k^3$–weighted $\chi(k)$ functions Fourier-transformed with Hanning windows of d$k$ = 1.0 Å$^{-1}$ to separate the EXAFS contributions from different coordination shells. A $k$-range of 2.8–11.3 Å$^{-1}$ was carried out for curve

fittings. The number of independent points is $N_{ipt} = 2\Delta k \times \Delta R/\pi = 2 \times (11.3 - 2.8) \times (3.1 - 1.0)/\pi = 11.4$. As can be seen in the FT-EXAFS curves of the ex situ $Pt = N_2 = Fe$ ABA sample, two prominent peaks at around 1.42 Å and 2.63 Å can be assigned to the Fe−N and Fe−N−Pt coordination. Thus, the fitting for ex situ $Pt = N_2 = Fe$ ABA sample was conducted with two separate Fe–N and Fe−N−Pt scattering paths. The significant coordination structure changes around Fe atoms occur at 0.95 and 0.90 V for the in situ samples. The first coordination peaks under 0.95 and 0.90 V conditions show an obviously increased strength compared to the KOH condition. Considering that the possible presence of oxygen-associated species adsorbing on the active metal-sites, an additional Fe−O path was included during the fitting. The Debye−Waller factors ($\sigma^2$) and bond length ($R$) were treated as adjustable parameters for all conditions in all paths, the energy shift ($\Delta E_O$) for different paths under the same condition was set equal to reduce the number of adjustable fitting parameters, and the coordination numbers ($N$) of Fe−N under in situ conditions were fixed with reference to the ex situ sample. Thus, the number of the adjustable parameters for the KOH and 1.05 V condition sample is $N_{para} = 6 < N_{ipt}$. And for the sample at 0.95 and 0.90 V, $N_{para} = 9 < N_{ipt}$. Using these strategies, all the resulted structural parameters are displayed in Supplementary Table 1−2, all generated $R$ factors are <0.01, indicating excellent quality.

## Data availability

The data that support the findings of this study are available within the article and its Supplementary Information. The source data are available from the corresponding authors upon reasonable request.

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

## Acknowledgements

This work was supported by the National Natural Science Foundation of China (Grants No. U1932212 (Q.L.), 12135012 (S.W.), U1932109 (W.C.), and 11875257 (Q.L.)) and the Natural Science Foundation of Anhui Province (Grants No. 2208085J01 (Q.L.) and 2208085QA28 (H.S.)).

## Author contributions

Q.L. and H.S. conceived the project. W.Z. and J.J. carried out the experiments. Q.L., H.S., W.Z., W.C., Y.L., M.L., F.Y., W.W. and S.W. analyzed the experimental data. The manuscript was written by W.Z., H.S. and Q.L. with contributions from all authors.

## Competing interests

The authors declare no competing interests.
