## [Peer Review File · Nature Communications]

Regulating the scaling relationship for high catalytic kinetics and selectivity of the oxygen reduction reactionReviewers' comments:

Reviewer #1 (Remarks to the Author):

The authors reported the preparation of PtFe dual metal-carbon composites and argued that the formation of Pt=N₂=Fe dual sites facilitated the radical breaking of O-O bond, hence bypassing the scaling limitation in ORR. There are a number of critical issues that are missed in this paper.

1. Based on the description of sample preparation in the experimental section, there is no control/driving force of the pairing between Fe and Pt. It is hard to believe that the dual-metal atomic configuration is the leading structure within the samples (Fig 4e). The authors basically made this argument based on two pieces of experimental evidence. One is TEM (Fig 1b). The boxes did show some pairing of atoms. However, the drawing of the boxes is arbitrary. In fact, one can observe the formation of a number of few-atom clusters. To confirm the Pt-Fe pairs, local EELS measurements/imaging are needed. The other evidence is EXAFS (Fig 1f and 1g). The authors argued that the Fe-Pt path length of 0.28 - 0.30 nm is corresponding to Pt-N₂-Fe. Note that this is also close to the distance of a direct Pt-Fe bond. In a prior study (Journal of Catalysis 255 (2008) 162–179) with PtFe alloy nanoparticles, the Pt-Fe length was estimated to be 0.283 nm. Thus, direct bonding of Pt-Fe and the formation of FePt alloy clusters/particles cannot be excluded. The authors need to compare the XAS data with that of PtFe alloy nanoparticles.

2. The ORR electrochemistry was carried out in alkaline media. As the composite contained Fe and N dopants, it was likely that the activity was actually due to FeN_x moieties as reported extensively in the literature, which are known to be efficient ORR catalysts in alkaline media (and could even outperform Pt/C). Pt-based catalysts are typically used for acidic ORR. The authors need to include data in acid for a meaningful comparison. In addition, as clusters are likely produced in the samples (see above in #1), their contributions need to be carefully analyzed (e.g., J. Phys. Chem. Lett. 2019, 10, 3, 460–467; Research 2020, 9167829).

Did the samples exhibit any butterfly feature in hydrogen adsorption/desorption?

Also, the mass activity of 2200 A g_{meta-l} at 0.85 V, while much better than that of Pt/C, is actually subpar as compared to leading results in the literature (Science, 2018, 362, 1276–1281; Research 2020, 9167829). This raises questions about the claim that the scaling relationship was overcome with the dual metal sites. If such a limitation was indeed removed, should one expect a much enhanced performance?

The K-L plots in Fig 1e are incorrect. The three potentials are in the diffusion-controlled region, whereas K-L analysis is for the kinetics-diffusion mixed region.

3. In the in situ XAS study, when the electrode potential was moved cathodically, the Pt L₃ edge intensity decreased while the Fe K edge intensity increased. This is inconsistent with the dual-site adsorption of oxygen to Fe and Pt (Fig 4), where the high electron affinity of oxygen would induce electron transfer from both metal centers.

4. Fig 4b, what is the broad band centered around 1300 cm⁻¹, which seemed to grow in intensity with decreasing potential. By the way, the panel e label is missing.

Reviewer #2 (Remarks to the Author):

This paper reports the preparation of carbon-supported Pt-Fe dual-site catalysts for the oxygen reduction reaction (ORR). The dual-site catalyst showed high ORR activity in alkaline electrolytes with a half-wave potential of 0.95 V (vs. RHE) with high 4-electron selectivity. By using ex situ/in situ

EXAFS and in situ FT-IR, the authors suggested a possible reason for the high ORR activity; the dual-site catalyst promotes the formation of M–O–O–M intermediates instead of *OOH intermediates, thereby potentially breaking the scaling relation.

The high ORR activity shown by the dual-site catalyst and the probing of reaction intermediates by in situ FT-IR is remarkable points of this paper. However, there are many questionable and ambiguous points that should be clearly addressed, particularly the formation of active dual sites. In its present form, this reviewer cannot recommend the publication of this paper.

1. Regarding the formation of Pt–Fe dual sites, many questions are arising. (1) Why was a pair of Pt and Fe chosen? What about Pt/Pt or Fe/Fe or combination of other elements? The choice of Pt and Fe should be rationalized. (2) Why is only the dual Pt–Fe dual site generated from the solution containing both Pt and Fe precursors? Pt–Pt or Fe–Fe site can also be generated. (3) For samples containing only Pt or Fe sites (Pt AMS and Fe AMS, respectively), why is only the single Pt (or Fe) site formed? Both the dual-site and single-site catalysts are prepared in the same way, except the precursor composition. Hence, for Pt AMS and Fe AMS catalysts, the formation of dual sites cannot be ruled out.
2. Page 3, "The meta-position amino functionalization of carbon nanoflakes with controllable size can controllably anchor Fe atoms adjacent to Pt atoms that have appropriate intermetallic atomic spacing for the dual-site mechanism design rule.": No structural evidence, such as IR and solid-state NMR, was provided for meta-position amino functionalization on carbon flakes. Even more, no structural characterization data were provided for N-functionalized carbon flakes and their intermediates. Hence, this claim is at best speculative without any evidence.
3. Page 5, "We note that it is a great challenge to synthesize atomically precise bimetal sites on such a high surface area. To achieve this goal, the precursors of Fe and Pt were added together with ethylene glycol to form an organic molecular metal solution, promoting the effective association between bimetals.": Why does the formation of organic molecular metal solutions promote the generation of bimetallic sites?
4. Page 5, It would be helpful to add a scheme for the catalyst preparation for readers' convenience.
5. Page 6, "...and the atomic concentrations of Pt and Fe in Pt=N2=Fe ABA determined by XPS are 3.8 and 1.4 wt%, respectively,...": atomic concentrations -> weight percentage
6. Page 6, "The N 1s XPS spectra are deconvoluted into a typical peak at 399.5 eV, which could be attributed to the N atom interaction with metal sites (Pt/Fe–Nx) in Pt=N2=Fe ABA (Supplementary Fig. 5).": 1) The corresponding XPS spectrum shows a peak at 399.0 eV, not 399.5 eV. 2) Supplementary Fig. 5 -> Supplementary Fig. 4
7. EXAFS curve fitting results of the dual-site catalyst reveal the position of the second peak at around 2.6 Å. But, TEM results show the inter-site distance of 2.8–3.0 Å. Why is the distance from the two analyses different? In EXAFS spectra, does R represent "reduced distance" instead of "distance"?
8. What about the ORR activity and selectivity of the prepared catalysts?

Reviewer #3 (Remarks to the Author):

The authors make and characterize a Pt-N2-Fe material that is reported to have a dual Fe-Pt active site for ORR. The authors boldly claim that their work "successfully breaks the limit of the inherent linear scaling relationship and causes the ORR on ABA to follow a fast-kinetic dual-site mechanism" and that "To the best of our knowledge, such a design principle for the above-mentioned dual-site mechanism has never been achieved in ORR electrocatalysis until now." I do not believe that the current manuscript has sufficient support of these lofty goals. Specifically, I am concerned that the Pt=N2=Fe material has more than just bimetallic Pt-Fe sites, that there is not an Fe AMS control for their SR-FTIR and that the *O-O* feature could be a different species, and there is no surface area normalization for their ORR RDE data making a clear activity comparison difficult.

Figure 1A-N: it looks like there could be other types of sites in the material – specifically isolated Fe and Pt sites or even small Pt/Fe nanoparticles that are not within the minimum distance for breaking scaling relations. Figure S2 also seems to indicate this. How is this accounted for and what would this

mean for the interpretation of the data? Approximately what fraction of the Fe-Pt sites are near each other?

Figure 1F-H: The Pt-Pt feature (from the Pt-metal) is very close to the feature noted with the pink dotted line. From EXAFS (as well as TEM-EDS) it seems possible that there are Pt nanoparticles in your sample. How different is the fit with a Pt-Pt path rather than an Fe-Pt path? Similarly, for the Fe K-edge, how different is the fit if a Fe-O path is used instead of a Pt-Fe path (I would expect Fe nanoparticles of such a small size to be fully oxidized). The average distance between Fe and Pt is a very important part of your analysis, however it is only a tiny part of the data. Please include a more rigorous assessment of why the Fe-Pt ~ 0.3 nm component is the best option for the data, not just an option. How reproducible is this EXAFS feature? Does it change with XAS scans or change substantially with minor normalization differences? Lastly, the Fe AMS and Pt AMS are used as single atom comparisons, however, is it possible that these too have nanoparticles?

Figure 2A: Almost -7 mA/cm² for a MT limited current density is higher than expected. Can you explain this?

For performance normalization it is standard to compare surface areas of the materials tested. How do the electrochemical, BET, or other surface areas compare? Is it possible that the Pt=N₂+Fe ABA just has the highest number of available active sites? It is also standard with Pt based materials to normalize based on the mg of Pt. How do these materials compare to each other and other PGMs on a mA/mgPt basis? Also, the TOF calculation was not clear. For the Pt=N₂=Fe is each pair considered an active site? How does the TOF change if each metal is considered?

Figure 4: Fe and Pt are known to have different ORR mechanisms. Because there are so few references in the literature for this *O-O* species it seems vital to have an Fe AMS control to make sure that the signal attributed to the O-O is not something that would have been seen for an Fe-only active site. Specifically, after looking at the literature it seems as though a "free superoxide species (O₂⁻), typically absorbs in the range 1,200–1,070 cm⁻¹ (10.1038/nchem.1874)"

Error bars from the RDE samples tested in triplicate (main text), as well as nitrogen CVs and the forward and backward sweep (SI) should be included to demonstrate that this is a reproducible performance and clarify the materials entire electrochemical profile.

Random:

AMS is not defined until after it shows up in Figure 1. If this material is breaking Pt scaling relations it seems like it should be more active than all Pt.

Please include a table of XPS fits and details from fits in the SI (e.g. peak positions, ratios, etc.)

Figure 1C: Based on the HAADF-STEM, what is the range in distance between the Fe and Pt?

The Pt/C control tested seems less active and stable than standard commercial Pt/C. How does this compare to literature values of commercial Pt?

Does the oxidation state fits from XPS and XAS match for the Pt and Fe? Please include quantitative analysis comparison.

Reply to Reviewer #1

The authors reported the preparation of PtFe dual metal-carbon composites and argued that the formation of Pt=N₂=Fe dual sites facilitated the radical breaking of O-O bond, hence bypassing the scaling limitation in ORR. There are a number of critical issues that are missed in this paper.

First of all, we greatly appreciate your comments and constructive suggestions on our manuscript. We have made a point-by-point response to your comments and carefully revised the manuscript as you suggested. For the synthesis of Pt-Fe dual-site that the reviewers mainly concerned about, we supplemented more experimental evidences on the basis of clear material preparation strategy. In addition, we added more details about the electrochemical performance evaluation to make the values more informative.

1. Question: Based on the description of sample preparation in the experimental section, there is no control/driving force of the pairing between Fe and Pt. It is hard to believe that the dual-metal atomic configuration is the leading structure within the samples (Fig 4e). The authors basically made this argument based on two pieces of experimental evidence. One is TEM (Fig 1b). The boxes did show some pairing of atoms. However, the drawing of the boxes is arbitrary. In fact, one can observe the formation of a number of few-atom clusters. To confirm the Pt-Fe pairs, local EELS measurements/imaging are needed. The other evidence is EXAFS (Fig 1f and 1g). The authors argued that the Fe-Pt path length of 0.28 - 0.30 nm is corresponding to Pt-N₂-Fe. Note that this is also close to the distance of a direct Pt-Fe bond. In a prior study (Journal of Catalysis 255 (2008) 162–179) with PtFe alloy nanoparticles, the Pt-Fe length was estimated to be 0.283 nm. Thus, direct bonding of Pt-Fe and the formation of FePt alloy clusters/particles cannot be excluded. The authors need to compare the XAS data with that of PtFe alloy nanoparticles.

Reply: Thank you for your nice questions. We are sorry that the synthesis and verification of atomic-scale dual-site pairs were not clearly addressed in the previous manuscript. These issues will be further discussed in detail in the revised manuscript.

For the synthesis of Pt=N₂=Fe ABA sample, the bimetallic precursor solution that facilitated the affinity of [PtCl₆]²⁻ to Fe³⁺ was prepared. The electronegative [PtCl₆]²⁻ ionized from H₂PtCl₆ and the electropositive Fe³⁺ ionized from FeCl₃ can promote effective attraction of them in ethylene glycol solvent. The attraction of two hetero-electric metal groups in organic solutions is considered to be a feasible strategy for the preparation of metal dual-site [*Angew. Chem. Int. Ed.* **2022**, 61, e2021161]. The precursor solution was mixed with functional amine-rich carbon nanoflakes to form atomic-level heteronuclear dual-site structure, which was confirmed by the atomic-resolved HAADF-STEM and XAS results.

Based on the Cs-corrected STEM-HAADF images of the Pt=N₂=Fe ABA sample, we performed a statistical analysis of the proportion of metal dual-site. Although the dual-site was not the only structure within the sample, it dominated about 72.6 % (Fig. N1, statistics on 500 randomly selected metal sites). And 25.3 % single metal sites and only 2.1% metal clusters were also contained in Pt=N₂=Fe ABA sample. To clarify the composition of the dual-site structure, as suggested by reviewers, electron energy-loss spectroscopy (EELS) measurement was employed (Fig. N2). We need to make an explanation to the results of the EELS measurement. EELS specializes in the detection of light elements with excellent energy collection ranges of 300-1000 eV, while the energy values of the electron excitation for Pt M_{4,5}-edge is 2122 eV. The analysis signal of the heavy elements in the high energy loss region is easily covered by the K-edge ionization loss peak of light elements, and the signal background that belongs to light elements is difficult to deduct. We have tried the signal acquisition of Pt and Fe elements in Pt=N₂=Fe ABA sample, as shown in Fig. N2. By scanning in a relatively large area (~5 × 5 nm), the results only show the signal peak of Fe element at 708.5 eV, and no obvious signal peak of Pt element was collected. Fig. N2b-c is the integral results of the red box area. The signal at 2122 eV is difficult to be assigned to a real element signal due to its poor signal-to-noise ratio. In this case, we distinguish the two metals (Pt and Fe) based on the commonly adopted theory that the contrast intensity of atoms in the Cs-corrected STEM-HAADF images highly depends on the atomic number (Z) [*Nat. Nanotechnol.* **2022**, 17, 590-602]. We supplemented the Cs-corrected STEM-

HAADF images with high-magnification acquired in the Pt=N₂=Fe ABA sample (Fig. N3), in which the dual-site structure can be clearly observed and accounts for the majority. Further, intensity analysis was performed at four dual-site regions of STEM-HAADF images (Fig. N3d-e and Fig. N4b). All the two bright spots along the arrow show different intensity, confirming that the dual-site is composed of Pt and Fe atoms. Thus, the Pt-Fe dual-site structure is the dominant form of metal sites, which has been confirmed repeatedly in different areas throughout the catalyst.

FT-EXAFS spectra of Pt *L*₃-edge and Fe K-edge were employed to study the coordination environment of the metal sites. For the Pt *L*₃-edge FT-EXAFS spectrum of Pt=N₂=Fe ABA sample in Fig. 1f of the manuscript, in addition to an obviously non-metallic coordination peak located at 1.55 Å, the metal coordination peak was observed at 2.63 Å, which is larger than the Pt-Pt coordination peak in Pt foil (located at 2.45 Å). The Fe K-edge FT-EXAFS spectrum of Pt=N₂=Fe ABA sample presents the metal coordination peak at 2.60 Å that is larger than the Fe-Fe coordination peak in Fe foil (located at 2.15 Å). According to the reviewer's constructive suggestion, we added the Pt *L*₃-edge FT-EXAFS spectrum of Pt-Fe alloy samples for a more careful comparison to exclude the presence of direct Pt-Fe coordination (Fig. N5). The strong Fourier transform peak ascribed to the scattering of Pt-Fe bond in the Pt-Fe alloy sample is located at 2.16 Å, and it is significantly lower than the location of metal coordination peak in Pt=N₂=Fe ABA sample. Furthermore, the information of the actual coordination bond length (*R*) and coordination numbers (CNs) was obtained by FT-EXAFS curve-fitting analysis. According to the best fitting results of ex situ Pt=N₂=Fe ABA sample (Table N1), the coordination bond length (*R*) of Pt-Fe (Fe-Pt) path was about 2.86 Å in both Pt *L*₃-edge and Fe K-edge EXAFS fitting, much longer than that of Pt-Fe alloy (< 2.65 Å) reported in the literature [*Nat. Comm.* **2014**, 5, 5634; *Nat. Catal.* **2022**, 5, 503-512]. For the typical study provided by reviewer about the PtFe compound adsorbs on SiO₂ surface [*Journal of Cat.* 2008, 255, 162–179], the material of PtFe(CO)₃(COD)[PhCC(H)C(H)CPh] belongs to the heteronuclear metal complex rather than metal alloy. Seen from the structural formula extracted from the literature

shown in Fig. N6, the metal bond length in metal complex is elongated due to the influence of the functional groups [*Organometallics* **1990**, 9, 2350-2357].

Accordingly, Fig. N4 **has been added** in revised manuscript as Fig. 1b-c. Fig. N1 and N3 **have been added** in Supplementary Information as Supplementary Fig. 7 and 8, respectively. In line 6, page 5 of the revised manuscript, the following text **has been added**: “For more heteronuclear metal dual-site to be synthesized, the precursor solution containing Fe^{3+} and $[PtCl_6]^{2-}$ was prepared in glycol solvent to promote the effective affinity between the two hetero-electric metal groups.” And in reverse line 1, page 5 of the revised manuscript, the following text **has been added**: “Statistical analysis based on 500 randomly selected metal sites reveals that the metal dual-site structure dominated about 72.6 % (Supplementary Fig. 7). Based on the fact that the intensity of atoms in the STEM-HAADF image highly depends on their atomic number (Z), intensity analysis was performed at four dual-site regions in STEM-HAADF images to clarify the composition of the dual-site structure (Fig. 1c and Supplementary Fig. 8d-e). All the two bright spots along the arrow show different intensity, confirming that the dual-site is composed of Pt and Fe atoms. The statistical analysis results also show that the atomic distance between Pt and Fe is approximately 2.81–2.93 Å.”

Fig. N1 The proportional distribution of the dual-site, single atom site and cluster in Pt=N₂=Fe ABA sample.

Fig. N2 EELS spectrum for the atomic sites shown in the red box in Cs-corrected STEM-HAADF images of Pt=N₂=Fe ABA.

Fig. N3 Characterization of Pt-Fe dual-site. **a** Cs-corrected HAADF-STEM images of Pt=N₂=Fe ABA. **b-c** Enlarged Cs-corrected HAADF-STEM images. **d-e** The intensity profile obtained from an individual dual-site.

Fig. N4 **a** Cs-corrected STEM-HAADF image of the Pt=N₂=Fe ABA catalyst. **b** The intensity profile obtained from an individual dual-site.

Fig. N5 Comparison of the FT-EXAFS spectra for Pt *L*₃-edge of Pt=N₂=Fe ABA sample.

Figure N6. ORTEP drawing of PtFe(CO)₃(COD)[M-PhCC(H)C-(H)CPh]. [*Organometallics* **1990**, *9*, 2350-2357]

Table N1 Structural parameters of the Pt *L*₃-edge and Fe *K*-edge for Pt=N₂=Fe ABA catalyst extracted from quantitative EXAFS curve-fittings using the ARTEMIS module of IFEFFIT.

Sample	Path	CNs	$R(\text{\AA})$	$\sigma^2 (10^{-3} \text{\AA}^2)$	ΔE_0 (eV)	R-factor
Pt L_{3} -edge ex situ	Pt-N	4.1±0.2	2.01±0.02	5.2±1.3	7.5	0.005
	Pt-Fe	1.1±0.2	2.86±0.02	7.6±1.5		
Fe K -edge ex situ	Fe-N	4.2±0.2	1.95±0.02	6.3±1.0	6.3	0.004
	Fe-Pt	1.0±0.2	2.85±0.02	8.1±1.5		

2. Question: The ORR electrochemistry was carried out in alkaline media. As the composite contained Fe and N dopants, it was likely that the activity was actually due to FeN_x moieties as reported extensively in the literature, which are known to be efficient ORR catalysts in alkaline media (and could even outperform Pt/C). Pt-based catalysts are typically used for acidic ORR. The authors need to include data in acid for a meaningful comparison. In addition, as clusters are likely produced in the samples (see above in #1), their contributions need to be carefully analyzed (e.g., J. Phys. Chem. Lett. 2019, 10, 3, 460–467; Research 2020, 9167829).

Reply: Thank you for your constructive suggestion that is quite useful for improving the quality of this work. We have tested the ORR performance of Pt=N₂=Fe ABA and the reference catalysts in acidic conditions to confirm the contribution of Pt-Fe dual-site contribution. The characterization results are shown in Fig. N7. The Pt=N₂=Fe ABA sample displays a better half-wave potential ($E_{1/2}$) than that of the standard Pt/C catalyst (0.861 and 0.845 V vs. RHE, respectively), and a kinetic current density (J_k) at 0.85 V_{RHE} twice higher than that of standard Pt/C (10.07 and 4.38 mA cm⁻², respectively). In contrast, the atomically monometallic sites catalysts (Pt AMS and Fe AMS) exhibit insufficient ORR electrochemical activity with inferior half-wave potential (0.81 and 0.75 V vs. RHE, respectively) and kinetic current density (1.12 and 0.36 mA cm⁻², respectively). It should be noted that we controlled the metal loading to ensure that the density of metal molar number in the three as-synthesized catalysts (Pt=N₂=Fe ABA, Pt AMS and Fe AMS) is approximately equal ($\sim 3.5 \times 10^{-7}$ mol mg⁻¹, calculated from ICP-OES results), which is helpful in excluding that the high activity

of the Pt=N₂=Fe ABA is caused by the superposition of the number of active sites. The excellent activity of Pt=N₂=Fe ABA catalyst in both alkaline and acidic electrolytes suggests the synergistic effect of the Pt-Fe dual-site in the catalytic ORR process. The limited catalytic activity of the isolated metal-nitrogen (M-N_x) coordinated species in the Pt AMS and Fe AMS catalysts may be blamed on the high energy barrier of the O-O bond cleavage due to the end-on adsorption state of the intermediates (*OOH and *O₂) on the simplicity active site [*Chem. Rev.* **2018**, 118, 2302-2312]. The Pt-Fe dual-site in the Pt=N₂=Fe ABA catalyst will be an effective structure to promote the O-O bond cleavage directly, so as to optimize ORR catalytic pathway.

As for the dispersion of the metal sites in the Pt=N₂=Fe ABA sample, as stated in the reply to question #1, the Cs-corrected STEM-HAADF and XAFS characterization show that the Pt-Fe dual-site structure accounts for the majority in the sample, while the clusters take up only a small proportion (about 2.1 %). Thus, Pt-Fe dual-site would be the main contributor to the catalytic activity. In addition, the in-situ XAFS and SR-FTIR characterization has observed the generation of the key intermediate O-O to form the Pt-O-O-Fe structure, which suggests the suppressed generation of *OOH by promoting the direct cleavage of *O-O* radicals, thereby accelerating the kinetics of the rate-determining step. However, in the literature listed by the reviewer, the regulation of the reaction mechanism by metal particles only achieves the optimization of the relationship between the binding energies of the multiple intermediates.

Accordingly, Fig. N7 **has been added** in Supplementary Information as Supplementary Fig. 20, and in reverse line 7, page 10 of the revised manuscript, the following text **has been added**: “*The Pt=N₂=Fe ABA catalyst also exhibits excellent performance in acidic electrolyte in comparison with the monometallic sites catalysts (Pt AMS and Fe AMS) and the standard commercial Pt/C (Supplementary Fig. 20), which suggests the synergistic effect of the dual-site in Pt=N₂=Fe ABA for catalyzing the ORR process.*”

Fig. N7 comparison of the ORR performance between Pt=N₂=Fe ABA sample and the reference catalysts in O₂-saturated 0.1 M HClO₄. Error bars represent the standard deviation from at least three independent measurements.

Did the samples exhibit any butterfly feature in hydrogen adsorption/desorption?

Reply: Thank you for your deep consideration. We performed cyclic voltammetry measurements for the synthesized platinum-based catalysts in the potential region of 0.1-1.5 V vs. RHE (Fig. N8) and found that no hydrogen/hydroxyl adsorption and desorption (known as “butterfly”) was observed. It is known that the monodispersed metal platinum does not present the hydrogen adsorption and desorption peak characteristic due to the absence of metallic crystal surface. [*Nat. Catal.* **2022**, 5, 503-512; *Nat. Catal.* **2022**, 5, 513-523]. As studied by Chen et al., in the series of samples with stepped Pt concentration (Fig. N9), only the sample containing a number of Pt nanoparticles shows apparent hydrogen adsorption/desorption voltammetric feature [*Research* **2020**, 9167829]. Thus, species adsorption/desorption-related butterfly feature is generally used in studies of crystalline platinum surfaces [*J. Phys. Chem. C* **2010**, 114, 8414–8422; *Electrochimica Acta* **2000**, 45, 4101-4115].

Accordingly, Fig. N8 **has been added** in Supplementary Information as Supplementary Fig. 15, and in line 3, page 8 of the revised manuscript, the following text **has been added**: “The cyclic voltammetry curves were recorded for the synthesized platinum-based catalysts (Pt=N₂=Fe ABA and Pt AMS) in N₂-saturated 0.1 M KOH electrolyte, which present no hydrogen adsorption and/or desorption characteristic due to the monoatomic dispersion of platinum in the synthesized catalysts (Supplementary Fig. 15).”

Fig. N8 Cyclic voltammetry curves of Pt=N₂=Fe ABA and Pt AMS catalysts recorded at the potential range of 0–1.2 V vs. RHE and 50 mV s⁻¹ scan rate in N₂-saturated 0.1 M KOH electrolyte.

Fig. N9 CV curves of the Pt-NC and PtCo-NC samples in 0.1 M HClO₄ at the scan rate of 50 mV s⁻¹. Note that only Pt-NC-4 and PtCo-NC-4 show apparent hydrogen adsorption/desorption voltammetric features, whereas the other samples display only featureless responses, consistent with the formation of only Pt isolated atoms/few-atom clusters in the latter, as compared to Pt NPs in Pt-NC-4 and PtCo-NC-4. (This figure is from *Research* **2020**, 9167829)

Also, the mass activity of 2200 A g_{meta}⁻¹ at 0.85 V, while much better than that of Pt/C, is actually subpar as compared to leading results in the literature (*Science*, 2018, 362, 1276–1281; *Research* 2020, 9167829). This raises questions about the claim that the scaling relationship was overcome with the dual metal sites. If such a limitation was indeed removed, should one expect a much enhanced performance?

Reply: Thank you for your nice question. We really compliment the reviewer for your expert knowledge in the field of electrochemical. According to your question, we re-

considered the characterization of mass activity parameters. It is inappropriate for us to vaguely calculate the mass activity of the Pt=N₂=Fe ABA catalyst that includes the contribution of all metals (Pt and Fe) at an arbitrarily selected potential, which may make the value lack comparability with similar materials in other literatures and affect the assessment of the intrinsic activity of platinum in the catalyst. Normatively, we recalculated the mass activity of the prepared Pt=N₂=Fe ABA catalyst was 14.1 A mg_{Pt}⁻¹ at 0.90 V vs. RHE (Fig. N7) with the Pt loading of 2.9 μg_{Pt} cm⁻² on rotating disk electrode (RDE) according to the standardized ORR mass activity calculation formula of MA=J_k/M_{Pt} (the kinetic current densities J_k, mA cm⁻²; the Pt loading density M_{Pt}, μg_{Pt} cm⁻²). The mass activity achieved in Pt=N₂=Fe ABA exceeds that of the best platinum-based catalyst reported previously (12.36 A mg_{Pt}⁻¹) [Science, **2018**, 362, 1276–128] and nearly doubles the outstanding reported mass activity value of ~4-7 A mg_{Pt}⁻¹ (Table N2).

The fast-kinetic dual-site mechanism regulates the intrinsic linear scaling relationship to effectively boost the ORR catalytic activity and selectivity, which was revealed by in-situ XAFS and SR-FTIR characterization. The intrinsic activity indexes, such as turnover frequency (TOF, 15.3 e⁻·site⁻¹·s⁻¹), mass activity (14.1 A mg_{Pt}⁻¹) and 4-electron pathway selectivity (99%), have been greatly improved compared with the reported platinum-based catalysts, which makes it a promising material for practical application to the energy device of zinc–air cells [Energy Environ. Mater. 2021, 4, 307–335].

Accordingly, Fig. N10 **has been added** in revised manuscript as Fig. 2d. And Table N2 **has been added** in Supplementary Information as Supplementary Table 7, and in line 12, page 9 of the revised manuscript, the following text **has been added**: “Pt=N₂=Fe ABA sample displays an ultrahigh TOF of 15.3 e⁻·site⁻¹·s⁻¹ and a great MA of 14.1 A mg_{Pt}⁻¹ at 0.90 V vs. RHE, 32 and 61 times better than that of commercial Pt/C reference (0.48 e⁻·site⁻¹·s⁻¹, 0.23 A mg_{Pt}⁻¹) and the high MA of platinum in Pt=N₂=Fe ABA exceeds most of the reported platinum-based catalyst (Supplementary Table 7).” The calculation formula of mass activity **has been added** in the **methods** section at the end of the revised manuscript: “The mass activity calculated by MA=J_k/M_{Pt} (the kinetic current densities J_k, mA cm⁻²; the Pt loading density M_{Pt}, μg_{Pt} cm⁻²)”

Fig. N10 Mass activity and turnover frequency of Pt=N₂=Fe ABA and reference catalysts.

Table N2 Comparison of RDE mass activity of Pt=N₂=Fe ABA with several high-performance catalysts from published works.

Catalyst	Mass activity A·mg ⁻¹ _{Pt} @0.9 V vs. RHE	Reference
Pt=N ₂ =Fe ABA	14.1	This work
LP@PF-2	12.36	Science 2018, 362, 1276–1281
Mo-Pt ₃ Ni/C	6.98	Science 2015, 348, 1230–1234
Pt ₃ Ni/C nanoframe	5.7	Science 2014, 343, 1339–1343
PtNiCo NW	4.2	Sci. Adv. 2017, 3, 1601705
sd-Pt ₈₄ Ni ₁₂ Co ₄	7.1	Nat. Catal. 2022, 5, 513–523
PtCo-NC-3	4.16	Research 2020, 9167829
J-PtNWs/C	13.6	Science 2016, 354, 1414–1419

The K-L plots in Fig 1e are incorrect. The three potentials are in the diffusion-controlled region, whereas K-L analysis is for the kinetics-diffusion mixed region.

Reply: Thank you for your careful consideration. The applicable potential interval of the Koutecky-Levich equation will be discussed in detail here to ensure the correctness of the selected characteristic potentials. The K-L equation, which applies to the diffusion-controlled region of the polarization curve, actually establishes the

relationship between limiting diffusion current and rotating disk electrode (RDE) speed. Based on the K-L equation, the electron transfer number (n) can be obtained by fitting the function curves of the limiting diffusion current density (J_L) to different rotational speeds. [A.J. Bard, et al. *Electrochemical methods: fundamentals and applications*. New York: wiley. **1980**, 2]

For the polarization curves of the prepared Pt=N₂=Fe ABA catalyst (Fig. N11a), the potential range from 0.85 to 1.0 V vs. RHE belongs to the kinetics-diffusion mixed region, including mass transfer, electrode surface reaction and diffusion-controlled. The region with potential below 0.85 V vs. RHE, which shows a horizontal diffusion platform, is the diffusion-controlled region. The curves shown in Fig. N11a were obtained by performing linear scan voltammetry measurement with rotating disk electrodes (RDE) at different rotational speeds. For a specific oxygen reduction reaction pathway (the electron transfer number (n) is constant), the limiting diffusion current density (J_L) is proportional to the rotational speed. According to the Koutecky-Levich equation:

$$\frac{1}{J} = \frac{1}{J_L} + \frac{1}{J_K} = \frac{1}{B\omega^{1/2}} + \frac{1}{J_K} \quad (1)$$

$$B = 0.62nFC_0D_0^{2/3}V^{-1/6} \quad (2)$$

At a certain potential, with $\omega^{-1/2}$ as the abscissa and corresponding J_L^{-1} as the ordinate, the slope obtained by linear fitting is B^{-1} (1). Since the Faraday constant (F), diffusion coefficient (D_0) and electrolyte viscosity (V) are constant, the electron transfer number can be calculated from Formula (2). [*Nat. Mater.* **2011**, 10, 780-786; *Adv. Sci.* **2021**, 8, 2100120] Accordingly, as shown in Fig.N11b, we selected six groups of data ($\omega^{-1/2}$, J_L^{-1}) at different potentials during diffusion-controlled region to plot, and obtained the electron transfer number ~ 3.98 by linear fitting. In addition, for the kinetics-diffusion mixed region (0.85-1.0 V vs. RHE), there is a linear relationship between potential and the logarithm of kinetic current density, which is expressed as Tafel slope analysis. [*Adv. Energy Mater.* **2018**, 8, 1800955]

To make the presentation clearer, the text in reverse line 10, page 9 of the revised manuscript **has been modified**: “*The ORR pathway was assessed via polarization*

curve measurement at different rotation rates. The electron transfer number (n) was calculated to be approximately 3.98 according to the Koutecky–Levich (K – L) equation applied in the diffusion-controlled region (inset of Fig. 2e), proving that the Pt=N₂=Fe ABA catalyst favours the 4e⁻ pathway.”

Fig. N11 Polarization plots at regular rotation rates (a) and the corresponding K-L plots in the range of 0.6 to 0.85 V (b) for Pt=N₂=Fe ABA.

3. Question: In the in situ XAS study, when the electrode potential was moved cathodically, the Pt L₃ edge intensity decreased while the Fe K edge intensity increased. This is inconsistent with the dual-site adsorption of oxygen to Fe and Pt (Fig 4), where the high electron affinity of oxygen would induce electron transfer from both metal centers.

Reply: Thank you for your nice question. As shown in Fig. N12, the white-line intensity of Pt L₃-edge XANES spectra undergoes a process of first rising and then falling, which represents the change of the oxidation state. And the white-line intensity of Fe K-edge XANES spectra changes similarly with potential hysteresis (Fig. N12). According to the in-situ characterization results of EXAFS spectra and SR-FTIR, we speculate that the potential hysteresis is due to the asynchronous adsorption of oxygen-related intermediate on the electronically coupled Pt=N₂=Fe dual-site. The adsorption of *O₂ on the dual-site is divided into two steps: the end-on pre-adsorption on the Pt site and then co-adsorption of *O-O* radical at dual-site, as shown in Figure N13.

In-situ XAS was employed to reveal the catalytic mechanism based on the structure evolution of active metal sites during the ORR process. For the in-situ XANES spectra

of Pt L_3 -edge (Fig. N12a-b), the intensity of the white-line peak evolves with applied potential, which corresponds to the occupation of the Pt $5d$ -band states [*Nat. Comm.* **2020**, 11, 1029]. The white-line intensity of the catalyst is increased when a potential of 1.05 V was applied, indicating the increase of Pt oxidation state. This might be due to the adsorption of some oxo-containing species ($*O_2$) on the metal Pt site. In contrast, the white-line intensities of the Fe K -edge XANES spectra under KOH and 1.05 V conditions remain unchanged, suggesting that the structure of metal Fe site is stable without chemical adsorption of reactant species in the electrolyte. With the application of oxygen reduction potentials (1.05 \rightarrow 0.90 V vs. RHE), the white-line intensity of Pt L_3 -edge gradually decreases, indicating a higher Pt $5d$ occupancy caused by the charge transfer from the nearby atoms to the Pt during the catalytic reaction process. During this potential region (1.05 \rightarrow 0.90 V vs. RHE), the white-line intensities of the Fe K -edge XANES spectra first increasing (1.05 \rightarrow 0.95 V vs. RHE) and then decreasing (0.95 \rightarrow 0.90 V vs. RHE), which is a similar change that lags behind the Pt site, indicating the coordination with oxygen at the Fe site closely follows the Pt site. The preferential adsorption of oxygen species on Pt site may be due to the asymmetric charge density distribution caused by the coupling between heteronuclear metal sites, which makes Pt site have a smaller adsorption energy barrier for oxygen-related intermediates than Fe site [*Sci. Adv.* 2022, 8, eabo0762]. Combined with *in situ* SR-FTIR measurement (Fig. 4a in manuscript), the $*O-O*$ intermediate was detected in the Pt-Fe dual-site contained catalyst during the catalytic ORR process. Based on the fact that bimetallic sites are prone to have co-adsorption of oxo-containing intermediates [*Nat. Catal.* **2021**, 4, 1012], we speculate that the Pt-Fe dual-site promotes the evolution of O_2 into $*O_2$ radical and accelerates the further cleavage of O-O bond, during which O_2 molecules are adsorbed on the Pt sites to form $*O_2$ intermediate first, then the distance between the adjacent Pt and Fe atoms is contracted under the driving voltage, and $*O-O*$ radical is co-adsorbed on Pt and Fe active center to form Pt-O-O-Fe structure. This catalytic process is consistent with the *in situ* XAS results of Fe site coordination with the oxygen closely adjacent to the Pt site.

Accordingly, in line 10, page 11 of the revised manuscript, the following text **has been added**: “Interestingly, for the Fe K-edge XANES (Fig. 3b-c), the white-line intensity changes similarly with potential hysteresis, indicating the coordination with oxygen at the Fe sites are closely adjacent to the Pt sites.”

Fig. N12 The Pt L_{3} -edge (a) and Fe K -edge (c) XANES spectra, and (b), (d) are the corresponding local magnifications of the white-line peaks.

Fig. N13 Schematic of the ORR mechanism on Pt=N₂=Fe dual-site within Pt=N₂=Fe ABA catalyst.

4. Question: Fig 4b, what is the broad band centered around 1300 cm⁻¹, which seemed to grow in intensity with decreasing potential. By the way, the panel e label is missing.

Reply: Thank you for your careful inspection. We do need to explain the fluctuation of the curves at 1300 cm⁻¹ to avoid unnecessary misunderstanding. The in-situ FTIR is adept at detecting the emerged species during reaction process. For the acquisition of in-situ FTIR data, we will pre-collect the signal of the carbon cloth with catalyst loaded as the background signal under ex situ condition, while the collected signal under in-situ potential-applied conditions will be processed to deduct the background signal to clearly reflect the change in signals of emerged species. The fluctuation at 1300 cm⁻¹

should be regarded as the signal-to-noise ratio of the curve after deducting the background signal, because it does not change obviously throughout the in-situ measurement process (0.95 \rightarrow 0.70 V vs. RHE).

In addition, we also reconfirmed that no ORR-associated species show FTIR vibrations at around 1300 cm^{-1} . In addition, the width of the species vibration detected at the Mid IR fingerprint region (500 – 1500 cm^{-1}) is generally $\sim 50 \text{ cm}^{-1}$, while the vibration width at 1300 cm^{-1} is close to 200 cm^{-1} and the vibration is not newly emerged at the reaction process [*Trends Anal. Chem.* **2010**, 29, 453-463], so it cannot be attributed to the IR vibration band of any intermediate species.

The text in line 9, page 13 of the revised manuscript **has been modified**: “As a reference, Fig. 4b shows the FTIR measurement results of the Pt AMS sample under the same conditions. A new IR absorption band appears around 965 cm^{-1} after the applied potential exceeds 0.95 V vs. RHE.” Moreover, thanks for your kind reminder, the e label has been added in Fig. 4 of the revised manuscript.

Reply to Reviewer #2:

This paper reports the preparation of carbon-supported Pt–Fe dual-site catalysts for the oxygen reduction reaction (ORR). The dual-site catalyst showed high ORR activity in alkaline electrolytes with a half-wave potential of 0.95 V (vs. RHE) with high 4-electron selectivity. By using ex situ/in situ EXAFS and in situ FT-IR, the authors suggested a possible reason for the high ORR activity; the dual-site catalyst promotes the formation of M–O–O–M intermediates instead of *OOH intermediates, thereby potentially breaking the scaling relation.

The high ORR activity shown by the dual-site catalyst and the probing of reaction intermediates by in situ FT-IR is remarkable points of this paper. However, there are many questionable and ambiguous points that should be clearly addressed, particularly the formation of active dual sites. In its present form, this reviewer cannot recommend the publication of this paper.

First of all, we greatly appreciate the reviewer's positive comments and constructive suggestions on our manuscript. We have made a point-by-point response to your comments and carefully revised the manuscript as you suggested. In detail, we have supplemented more information about the material preparation strategy and experimental evidence for the issues of the Pt-Fe dual-site formation that reviewers concerned about. Our replies to your questions and criticisms are as follows.

1. Question: Regarding the formation of Pt–Fe dual sites, many questions are arising. (1) Why was a pair of Pt and Fe chosen? What about Pt/Pt or Fe/Fe or combination of other elements? The choice of Pt and Fe should be rationalized. (2) Why is only the dual Pt–Fe dual site generated from the solution containing both Pt and Fe precursors? Pt–Pt or Fe–Fe site can also be generated. (3) For samples containing only Pt or Fe sites (Pt AMS and Fe AMS, respectively), why is only the single Pt (or Fe) site formed? Both the dual-site and single-site catalysts are prepared in the same way, except the precursor composition. Hence, for Pt AMS and Fe AMS catalysts, the formation of dual sites cannot be ruled out.

Reply: Thank you for your nice question. We will make a point-by-point response to your comments in the following.

(1) Compared with the conventional isolated single-atom sites catalysts, the correlated metal sites catalysts (named C-SACs) have been proven to induce evident electronic effects via electron transfer between adjacent metal single atoms (*J. Am. Chem. Soc.* **2021**, 143, 16068–16077; *ACS Catal.* **2020**, 10, 7584–7618). And the heteronuclear metal site (M/M') may potentially exhibits structural deformation and asymmetric charge density, which proves to be more flexible in tuning electronic and electrocatalytic properties than the homonuclear metal site [*J. Catal.* **2020**, 388, 77–83]. In principle, the pairing and coupling of different single-atom would induce charge polarization, resulting in the difference of binding energy for the intermediate at the two paired sites, which is beneficial for the activity in complicated reactions involving multistep proton-electron transfer [*Sci. Adv.* **2022**, 8, eabo0762]. Therefore, in this work, based on the synergistic effect and the difference in binding energies for oxygen-intermediates of the heteronuclear metal sites (Pt and Fe) [*Nat. Comm.* **2021**, 12, 3252], in-situ XAFS and FTIR characterization proves that the *O₂ intermediate was pre-adsorbed on Pt site, and then promotes the direct cleavage of *O-O* radical in the form of Pt-O-O-Fe dual-site adsorption, which is an effective way to accelerate the kinetics of the rate-determination step in ORR process.

On the other hand, for the selection of heteronuclear metal species, we also consider the catalyst activity and practical application status. Pt-based electrocatalysts are recognized as the best performing catalyst for ORR [*Nat. Nanotechnol.* **2016**, 11, 1020]. However, the Pt nanoparticles catalysts with high-content of Pt are mainly used for fuel cells in the market, which are of high cost [*J. Phys. Chem. Lett.* **2016**, 7, 1127–1137]. In recent years, iron-based catalysts (represented by FeN₄ structure) have been proved by increasing studies to be the most potential transition metal based catalysts for catalytic ORR, but the industrial application seems to be limited by the poor stability of the metal sites [*Energy Environ. Mater.* **2021**, 4, 307–335]. Therefore, replacing part of platinum with active transition metal seems to be an effective strategy to reduce the cost while improving the activity and stability for commercial application, which has

led to extensive research on commercial electrocatalysts with Pt and Fe as co-active metals.

(2) In order to promote the formation of more Pt-Fe dual-site, the bimetallic precursor solution that facilitated the affinity of $[\text{PtCl}_6]^{2-}$ to Fe^{3+} was prepared. The electronegative $[\text{PtCl}_6]^{2-}$ ionized from H_2PtCl_6 and the electropositive Fe^{3+} ionized from FeCl_3 can promote effective attraction of them in ethylene glycol solvent. The attraction of two hetero-electric metal groups in organic solvents is considered to be a feasible strategy for the preparation of metal dual-site [*Angew. Chem. Int. Ed.* **2022**, 61, e2021161]. Indeed, this strategy does not guarantee that all formed metal sites are the structure of Pt-Fe atom pairs in the sample. According to the Cs-corrected STEM-HAADF images of the as-synthesized Pt=N₂=Fe ABA sample in Fig. N14-15, the dual-site structure accounts for the main part. Further, intensity analysis was performed at four dual-site regions of STEM-HAADF images (Fig. N14d-e and Fig. N15b). All the two bright spots along the arrow show different intensity, confirming that the dual-site is composed of Pt and Fe atoms. In addition, we conducted statistics on 500 randomly selected metal sites in the Pt=N₂=Fe ABA sample based on the Cs-corrected STEM-HAADF images [*Adv. Mater.* **2021**, 33, 2003327]. As shown in Figure N16, the dual-site structure dominates about 72.6 %. And the in-situ characterization reveals that the dual-atom structure is the main contribution for catalytic performance.

(3) As described in reply to #(2), the formation of the dual-site structure depends on the attraction of $[\text{PtCl}_6]^{2-}$ and Fe^{3+} in organic solvents due to their opposite electrical properties. For the synthesis of single-site catalysts (Pt AMS and Fe AMS), the separate aqueous solution of H_2PtCl_6 or $\text{FeCl}_3 \cdot 6\text{H}_2\text{O}$ is mixed directly with the prefabricated ammonia-rich carrier (CNF-NH₂). The homoelectric metal ions tend to disperse uniformly in the CNF-NH₂ solution and do not have the conditions to form the dual-atom structure .

Accordingly, Fig. N14, 16 **have been added** in Supplementary Information as Supplementary Fig. 8 and 7, respectively. Fig. N15 **has been added** in revised manuscript as Fig. 1b-c. In line 6, page 5 of the revised manuscript, the following text **has been added**: “For more heteronuclear metal dual-site to be synthesized, the

precursor solution containing Fe^{3+} and $[PtCl_6]^{2-}$ was prepared in glycol solvent to promote the effective affinity between the two hetero-electric metal groups.” And in reverse line 1, page 5 of the revised manuscript, the following text **has been added:** “Statistical analysis based on 500 randomly selected metal sites reveals that the metal dual-site structure dominated about 72.6 % (Supplementary Fig. 7). Based on the fact that the intensity of atoms in the STEM-HAADF image highly depends on their atomic number (Z), intensity analysis was performed at four dual-site regions in STEM-HAADF images to clarify the composition of the dual-site structure (Fig. 1c and Supplementary Fig. 8d-e). All the two bright spots along the arrow show different intensity, confirming that the dual-site is composed of Pt and Fe atoms. The statistical analysis results also show that the atomic distance between Pt and Fe is approximately 2.81–2.93 Å.” And in the “Methods” section at the end of the revised manuscript, **the preparation process was supplemented in detail:** “Typically, a mixed solution was made by adding 5.75 mg of H_2PtCl_6 and 2.97 mg of $FeCl_3$ to the glycol solvent. The mixture was ultrasonicated and mechanical stirred for 20 min to promote the effective association between electropositive Fe^{3+} and electronegative $[PtCl_6]^{2-}$.”

Fig. N14 Characterization of Pt-Fe dual-site. **a** Cs-corrected HAADF-STEM images of Pt=N₂=Fe ABA. **b-c** Enlarged Cs-corrected HAADF-STEM images. **d-e** The intensity profile obtained from an individual dual-site.

Fig. N15 **a** Cs-corrected STEM-HAADF image of the Pt=N₂=Fe ABA catalyst. **b** The intensity profile obtained from an individual dual-site.

Fig. N16 The proportional distribution of the dual-site, single atom sites and cluster in Pt=N₂=Fe ABA sample..

2. Question: Page 3, “The meta-position amino functionalization of carbon nanoflakes with controllable size can controllably anchor Fe atoms adjacent to Pt atoms that have appropriate intermetallic atomic spacing for the dual-site mechanism design rule.”: No structural evidence, such as IR and solid-state NMR, was provided for meta-position amino functionalization on carbon flakes. Even more, no structural characterization data were provided for N-functionalized carbon flakes and their intermediates. Hence, this claim is at best speculative without any evidence.

Reply: Thank you for your kind reminder that is very important to improve the integrity of this work. The preparation route of the catalyst involves many intermediates, and the structural characterizations of these intermediates are necessary. For the preparation of the support material, pyrene as the raw material is firstly nitrated to nitropyrene (denoted as CNF-NO₂), which is easy to induce substitution reaction in alkaline solution, and then functionalized by amino group in aqueous ammonia to form CNF-NH₂. The CNF-NH₂ was mixed with the metal precursor solution and freeze-dried to form metal-loaded CNF-NH₂ (denoted as M/CNF-NO₂). Followed by a pyrolysis process under an ammonia-rich atmosphere, the atomic-scale bimetal assembly (Pt=N₂=Fe ABA) catalyst can be obtained. The preparation scheme is shown in Fig. N20. During the synthesis process, the nitro-functionalized intermediate of trinitropyrene (CNF-NO₂) and the further amino-functionalized carbon nanoflakes (CNF-NH₂) have been certified by X-ray photoelectron spectroscopy (XPS) characterization in the revised manuscript. Structural characterization of CNF-NH₂ and its metal-loaded M/CNF-NH₂ sample (un-annealed) by FTIR has been supplemented to demonstrate the binding of the metal to the amino functionalized points. Moreover, the added N *K*-edge XANES spectra of carbonized CNF-NH₂ and Pt=N₂=Fe ABA (M/CNF-NH₂ annealed at 700 °C) reveal the coordination of anchored metal atoms with amino-derived nitrogen atoms in Pt=N₂=Fe ABA catalyst.

The high-resolution N 1s XPS spectrum (Fig. N17a) of the CNF-NO₂ intermediate displays the strong signal of N-O at 405.5 eV, which represents the main presence of -NO₂ in the sample [*Nat. Comm.* **2014**, 5, 5357]. The high-resolution N 1s XPS spectrum of the CNF-NH₂ can be fitted to the signals of N-H at 399.7 eV and N-C at 401.5 eV, respectively (Fig. N17b). There is only a weak signal that represents the residual -NO₂ at 406.3 eV. The XPS results reveal that the amino group (-NH₂) almost completely replaced the nitro group (-NO₂) in the amino functionalization step to obtain the amino-functionalized carbon nanoflakes (CNF-NH₂). The samples before and after metal loading (CNF-NH₂ and M/CNF-NH₂) were characterized and analyzed by FTIR (Fig. N18). The FTIR spectrum of M/CNF-NH₂ shows the retention of chemical structures in the CNF-NH₂, except for the apparent red shift and broadening of the ν_{N-C} peak,

indicating the coordination of metal ions to the N atoms of the amino group [*Chem. Mater.* **2016**, 28, 4375–4379; *Nat. Comm.* **2014**, 5, 5357]. The N *K*-edge XANES spectra of the Pt=N₂=Fe ABA exhibits the coordination peak of the metal sites (Pt/Fe) with the nitrogen atoms (Fig. N19). And the enhanced N-C π^* peak at 293.3 eV and broadened σ^* peak at ~593 eV could be attributed to the covalent coupling between metal and nitrogen, confirming the strong interaction between metal and nitrogenous matrix in the Pt=N₂=Fe ABA [*J. Am. Chem. Soc.* **2012**, 134, 3517–3523].

Accordingly, Fig. N17, N18 and N19 **have been added** in Supplementary Information as Supplementary Fig. 4, 5 and 6. In line 4, page 5 of the revised manuscript, the following text **has been added**: “*The amino-functionalized carbon nanoflakes (CNF-NH₂) was obtained after the nitro group (-NO₂) almost completely replaced by the amino group (-NH₂) in amino functionalization step (Supplementary Fig. 4).*” And in line 13, page 5 of the revised manuscript, the following text “*The Fourier transform infrared spectroscopy (FTIR) characterization results (Supplementary Fig. 5) reveal that the metal was successfully modified on the amino group of CNF-NH₂. And after annealing at 700 °C, the metal remains the strong interaction with nitrogenous matrix (Supplementary Fig. 6).*” **has been added**.

Fig. 17 a N 1s XPS spectrum of CNF-NO₂ **b** The fitted of N 1s XPS spectrum of CNF-NH₂. The strong signal of N-O at 405.5 eV in (a) represent the main presence of -NO₂ in the CNF-NO₂ sample. In contrast, the fitting results of N1s XPS spectrum for CNF-NH₂ represent the main signals of N-H at 399.7 eV and N-C at 401.5 eV, respectively, and the weak signal of residual -NO₂ at 406.3 eV. It reveals that the amino group (-NH₂) almost completely replaces the nitro group (-NO₂) in the amino functionalization step to obtain the amino-functionalized carbon nanoflakes (CNF-NH₂).

Fig. N18 Comparative FTIR spectra of CNF-NH₂ before and after metal modification. The chemical structures are retained except for the apparent redshift and widening of the $\nu_{\text{N-C}}$ peak at $\sim 1120 \text{ cm}^{-1}$, indicating the coordination of metal ions to the N atoms of the amino group.

Fig. N19 N K-edge XANES of Carbonized CNF-NH₂ (CNF-NH₂ annealed at 700 °C) and Pt=N₂=Fe ABA (M/CNF-NH₂ annealed at 700 °C). The coordination peak of metal sites (Pt/Fe) with nitrogen atoms can be clearly observed at 299.4 eV. And the enhanced N-C π^* peak at 398.1 eV and broadened σ^* peak at $\sim 408.1 \text{ eV}$ of Pt=N₂=Fe ABA attributed to the covalent coupling between metal and nitrogen, confirming the strong interaction between metal and nitrogenous matrix.

3. Question: Page 5, “We note that it is a great challenge to synthesize atomically precise bimetal sites on such a high surface area. To achieve this goal, the precursors of Fe and Pt were added together with ethylene glycol to form an organic molecular metal solution, promoting the effective association between bimetals.”: Why does the

formation of organic molecular metal solutions promote the generation of bimetallic sites?

Reply: Thank you for your nice question. We will address this issue in detail. The attraction of two hetero-electric metal groups in organic solvents is considered to be a feasible strategy for the preparation of metal dual-site [*Angew. Chem. Int. Ed.* **2022**, 61, e2021161]. Thus, in this work, the electronegative $[\text{PtCl}_6]^{2-}$ ionized from H_2PtCl_6 and the electropositive Fe^{3+} ionized from FeCl_3 can promote effective attraction of them in ethylene glycol solvent. The metal sites in the as-synthesized Pt=N₂=Fe ABA sample are performed atomic-resolution characterization, as stated in reply to question #1, which confirms that the dual-site composed of Pt and Fe is the dominant form.

We are aware that the description pointed out by the reviewer is ambiguous, so we have **made the following modifications** in line 6, page 5 of the revised manuscript: *“For more heteronuclear metal dual-site to be synthesized, the precursor solution containing Fe^{3+} and $[\text{PtCl}_6]^{2-}$ was prepared in glycol solvent to promote the effective affinity between the two hetero-electric metal groups.”*

4. Question: Page 5, It would be helpful to add a scheme for the catalyst preparation for readers' convenience.

Reply: Thank you for your constructive suggestion that is quite useful for improving the quality of this work. According to your nice suggestion, we drew the catalyst preparation process into a scheme to make the route clearer in Fig. N20. And Fig. N20 **has been added** in Supplementary Information as Supplementary Fig. 1. In reverse line 1, page 4 of the revised manuscript, the following text **has been added:** *“The preparation scheme can be seen in Supplementary Fig. 1.”*

Fig. N20 Schematic illustration for the synthetic procedure of Pt=N₂=Fe ABA.

5. Question: Page 6, “...and the atomic concentrations of Pt and Fe in Pt=N₂=Fe ABA determined by XPS are 3.8 and 1.4 wt%, respectively,...”: atomic concentrations -> weight percentage

Reply: Thank you for your careful inspection, and we have checked the issue of metal concentrations in the revision. We actually calculated the mass percentage of metals in the samples according to both XPS and ICP-OES results, and it was an oversight in our way of expressing, in which we miswrote “the metal concentrations of Pt and Fe” as “the atomic concentrations of Pt and Fe” in the text.

Accordingly, in line 10, page 6, the text “...and the atomic concentrations of Pt and Fe in Pt=N₂=Fe ABA determined by XPS are 2.93 and 1.34 wt%, respectively,” **has been changed to:** “the metal concentrations of Pt and Fe in Pt=N₂=Fe ABA determined by XPS are 2.93 and 1.34 wt%, respectively, which are close to the inductively coupled plasma–optical emission spectrometry (ICP–OES) results (inset of Supplementary Fig. 9).”

6. Question: Page 6, “The N 1s XPS spectra are deconvoluted into a typical peak at 399.5 eV, which could be attributed to the N atom interaction with metal sites (Pt/Fe–N_x) in Pt=N₂=Fe ABA (Supplementary Fig. 5).”: 1) The corresponding XPS spectrum

shows a peak at 399.0 eV, not 399.5 eV. 2) Supplementary Fig. 5 -> Supplementary Fig. 4

Reply: Thank you for your careful inspection. To make the description of the peak position more accurate, we re-examined the N 1s XPS fitting information. The characteristic peak of M-N_x in Fig. N21 is precisely located at 399.35 eV. In addition, we added a table containing XPS fitting information to improve the accuracy and completeness of the data (Table N3).

Fig. N21, Table N3 **have been added** in Supplementary Information as Supplementary Fig. 10 and Supplementary Table 1, respectively. And in line 14, page 6 of the revised manuscript, the following text **has been added**: “*The N 1s XPS spectra were fitted into a typical peak at 399.35 eV, which could be attributed to the N atom interaction with metal sites (Pt/Fe-N_x) in Pt=N₂=Fe ABA (Supplementary Fig. 8, Supplementary Table 1).*”

Fig. N21 The fitted N1s XPS spectra of Pt=N₂=Fe ABA sample.

Table N3 Fitting parameters of N 1s XPS spectra for Pt=N₂=Fe ABA sample.

Fitting of N 1s	Position (eV)	FWHM (eV)	Area
Pyridinic N	398.26	1.32	14640.93
Pt/Fe-N	399.35	2.58	15931.34
Graphitic N	400.78	1.82	20752.82
Oxidized N	403.01	3.60	4900.56

7. Question: EXAFS curve fitting results of the dual-site catalyst reveal the position of the second peak at around 2.6 Å. But, TEM results show the inter-site distance of 2.8-3.0 Å. Why is the distance from the two analyses different? In EXAFS spectra, does R represent “reduced distance” instead of “distance”?

Reply: Thank you for your nice question. Accurately, the location of the characteristic peak in the FT-EXAFS spectra does not represent the actual coordination bond length. The position of characteristic peak reflected by the radial function can roughly represent the distance between the central atom and the scattering atom, which is about 0.2-0.3 Å deviation from the actual atomic distance. The deviation is mainly due to the existence of single/multiple scattering that will produce the meditative phase difference. Both the central atom phase shift and the backscattering phase shift results in the difference between the radial function peak and the true atomic distance. In fact, the coordination numbers (CNs), actual coordination bond length (R) and other structural information can be exactly obtained by fitting FT-EXAFS spectra. The fitting results of the FT-EXAFS spectra in Figure N22a (Figure 1h in the revised manuscript) are shown in Table N4. It can be seen that the peak located at around 2.6 Å were fitted as the coordination between Pt and Fe, and the coordination bond length are around 2.86 Å, corresponding to the *atomic spacing analysis* results in Cs-corrected STEM-HAADF images (Figure N14 and 15).

Accordingly, in reverse line 10, page 7 of the revised manuscript, the following text **has been added:** “*The fitting results of both Pt L₃-edge and Fe K-edge FT-EXAFS spectra show that the bond length (R) of the metal coordination is around 2.86 Å, corresponding to the atomic spacing analysis results in Cs-corrected STEM-HAADF images.*”

Fig. N22 a FT-EXAFS spectra of Pt L_3 -edge and Fe K -edge, and the corresponding fitting curves for Pt=N₂=Fe ABA catalyst. **b** Intensity distribution along arrow direction characterized by Cs-corrected scanning transmission electron microscopy

Table N4 Structural parameters of the Pt L_3 -edge and Fe K -edge for Pt=N₂=Fe ABA catalyst extracted from quantitative EXAFS curve-fitting using the ARTEMIS module of IFEFFIT.

Sample	Path	CNs	$R(\text{Å})$	$\sigma^2 (10^{-3} \text{Å}^2)$	ΔE_0 (eV)	R-factor
Pt L_3 -edge ex situ	Pt-N	4.1±0.2	2.01±0.02	5.2±1.3	7.5	0.005
	Pt-Fe	1.1±0.2	2.86±0.02	7.6±1.5		
Fe K -edge ex situ	Fe-N	4.2±0.2	1.95±0.02	6.3±1.0	6.3	0.004
	Fe-Pt	1.0±0.2	2.85±0.02	8.1±1.5		

8. Question: What about the ORR activity and selectivity of the prepared catalysts?

Reply: Thank you for your nice question. Important results of the performance tests related to ORR activity and selectivity of the prepared samples are shown in Fig. N23. The linear sweep voltammetry (LSV) curves in Fig. 23a show that Pt=N₂=Fe ABA has a superior half-wave potential ($E_{1/2}$) of 0.95 V vs. RHE, which is 90 mV vs. RHE higher than that of commercial Pt/C (0.86 V vs. RHE) and also outperforming those of Pt AMS (0.83 V vs. RHE), Fe AMS (0.87 V vs. RHE). And the kinetic current density (J_k) of Pt=N₂=Fe ABA at 0.95 V vs. RHE (5.83 mA cm⁻²) is nearly two orders of magnitude higher than that of commercial Pt/C (J_k , 0.09 mA cm⁻²). The mass activity of the prepared Pt=N₂=Fe ABA catalyst was 14.1 A mg_{Pt}⁻¹ at 0.90 V vs. RHE, which nearly doubles the outstanding reported mass activity value of ~4-7 A mg_{Pt}⁻¹ in Platinum-based

catalyst (Fig.N23b). In the meanwhile, the as-prepared Pt AMS catalyst shows mass activity comparable to standard commercial Pt/C, which are 0.2 and 0.23 A mg_{Pt}⁻¹, respectively. The calculation results of turnover frequency (TOF) show that the dual-site catalyst (Pt=N₂=Fe ABA) has efficient active site conversion rate with 15.3 e⁻·site⁻¹·s⁻¹ (Fig.N23b). And the turnover frequency of the active site in Pt AMS (0.37 e⁻·site⁻¹·s⁻¹) and Fe AMS (3.2 e⁻·site⁻¹·s⁻¹) catalyst are at the same level with standard commercial Pt/C (0.48 e⁻·site⁻¹·s⁻¹). The selectivity for the 4-electron ORR pathway was assessed via polarization curve measurements on ring-disk electrode (RDE) at different rotation rates (Fig.N23c) and rotating ring-disk electrode (RRDE) (Fig. N23d). The electron transfer number (*n*) was calculated to be approximately 3.98 of Pt=N₂=Fe ABA catalyst according to the Koutecky–Levich (K–L) equation (inset of Fig. N23c), which is consistent with the measurement result in RRDE, proving that the Pt=N₂=Fe ABA catalyst favours the 4e⁻ pathway with the H₂O₂ yield below 1.5%. The selectivity for ORR pathway of the reference catalysts is also shown in Fig. 7d, with the electron transfer number between 3.9-3.95 and the H₂O₂ yield around 6%.

Accordingly, the measurement results of Fig. N23 have been shown in the revised manuscript as a part of Fig. 2, and the interpretation of the ORR activity and selectivity of the prepared samples, as stated above, is shown in the description section of Figure 2 in revised manuscript.

Fig. N23 **a** ORR polarization plots of Pt=N₂=Fe ABA and reference samples. **b** The turnover frequency and mass activities of the Pt=N₂=Fe ABA and reference samples at 0.9 V vs. RHE. (The standard of mass activity for Fe AMS sample is A mg_{Fe}⁻¹.) **c** Polarization plots at regular rotation rates for Pt=N₂=Fe ABA with the inset showing the derived K–L plots. **d** ORR pathway selectivity parameters for the catalysts involved.

Reply to Reviewer #3:

The authors make and characterize a Pt-N₂-Fe material that is reported to have a dual Fe-Pt active site for ORR. The authors boldly claim that their work “successfully breaks the limit of the inherent linear scaling relationship and causes the ORR on ABA to follow a fast-kinetic dual-site mechanism” and that “To the best of our knowledge, such a design principle for the above-mentioned dual-site mechanism has never been achieved in ORR electrocatalysis until now.” I do not believe that the current manuscript has sufficient support of these lofty goals. Specifically, I am concerned that the Pt=N₂=Fe material has more than just bimetallic Pt-Fe sites, that there is not an Fe AMS control for their SR-FTIR and that the *O-O* feature could be a different species, and there is no surface area normalization for their ORR RDE data making a clear activity comparison difficult.

First of all, we greatly appreciate your comments and constructive suggestions on our manuscript. Under your kind reminder, we realized that these expressions in the previous manuscript were indeed inappropriate. After adding more experimental evidences, we have made necessary deletions and/or improvements on these expressions of the manuscript. In addition, we have added more details of the electrochemical performance evaluation to make the values more comparative. The specific modifications are shown in point-to-point manner as follows.

1. Question: Figure 1A-N: it looks like there could be other types of sites in the material – specifically isolated Fe and Pt sites or even small Pt/Fe nanoparticles that are not within the minimum distance for breaking scaling relations. Figure S2 also seems to indicate this. How is this accounted for and what would this mean for the interpretation of the data? Approximately what fraction of the Fe-Pt sites are near each other?

Reply: Thank you for your nice question. As concerned by the reviewer, the synthetic strategy employed in this work does not guarantee that all metal sites are the structure of Pt-Fe dual-site in the synthesized samples. According to the Cs-corrected STEM-HAADF images of the as-synthesized Pt=N₂=Fe ABA sample in Fig. N24-25, the dual-

site structure accounts for the majority, and the atomic spacing in the dual-site is about 0.85 Å. Further, we conducted statistical analysis on 500 randomly selected metal sites in the Pt=N₂=Fe ABA sample based on the Cs-corrected STEM-HAADF images [*Adv. Mater.* **2021**, 33, 2003327]. As shown in Fig. N26, the dual-site structure dominates about 72.6%. In addition, ORR performance tests and the in-situ characterization results show that the activity of dual-site structure in Pt=N₂=Fe ABA sample is obviously better than the single-site reference catalyst.

According to the reviewer's comments, we have modified the inappropriate expressions of previous manuscript and added some meaningful experimental evidences in the revised manuscript. In reverse line 1, page 5 of the revised manuscript, the following text **has been added**: *“Statistical analysis based on 500 randomly selected metal sites reveals that the metal dual-site structure dominated about 72.6% (Supplementary Fig. 7). Based on the fact that the intensity of atoms in the STEM-HAADF image highly depends on their atomic number (Z), intensity analysis was performed at four dual-site regions in STEM-HAADF images to clarify the composition of the dual-site structure (Fig. 1c and Supplementary Fig. 8d-e). All the two bright spots along the arrow show different intensity, confirming that the dual-site is composed of Pt and Fe atoms. The statistical analysis results also show that the atomic distance between Pt and Fe is approximately 2.81–2.93 Å.”*

Fig. N24 Characterizations of Pt-Fe dual-atom site. **a** Cs-corrected HAADF-STEM images of Pt=N₂=Fe ABA. **b-c** Enlarged Cs-corrected HAADF-STEM images located at different positions. **d-e** The intensity profile obtained on one individual dual-atom site.

Fig. N25 a Cs-corrected STEM-HAADF images of the Pt=N₂=Fe ABA catalyst. **b** The intensity profile obtained on individual dual-atom site.

Fig. N26 The proportional distribution of the dual-site, single atom site and cluster in Pt=N₂=Fe ABA sample.

2. Question: Figure 1F-H: The Pt-Pt feature (from the Pt-metal) is very close to the feature noted with the pink dotted line. From EXAFS (as well as TEM-EDS) it seems possible that there are Pt nanoparticles in your sample. How different is the fit with a Pt-Pt path rather than an Fe-Pt path? Similarly, for the Fe K-edge, how different is the fit if a Fe-O path is used instead of a Pt-Fe path (I would expect Fe nanoparticles of such a small size to be fully oxidized). The average distance between Fe and Pt is a very important part of your analysis, however it is only a tiny part of the data. Please include a more rigorous assessment of why the Fe-Pt ~ 0.3 nm component is the best option for the data, not just an option. How reproducible is this EXAFS feature? Does it change with XAS scans or change substantially with minor normalization differences? Lastly, the Fe AMS and Pt AMS are used as single atom comparisons, however, is it possible that these too have nanoparticles?

Reply: Thank you for your nice question. We will make a point-by-point response to your comments. As concerned by the reviewer, the location of the metal coordination peak in the Pt L₃-edge EXAFS spectrum of Pt=N₂=Fe ABA sample is very close to that in Pt foil. Therefore, we carefully judged whether the sample contains metal nanoparticles. In addition to the absence of metal diffraction peaks in XRD characterization, atomic-level Cs-corrected STEM-HAADF characterization is a good mean. In the Cs-corrected STEM-HAADF images, we observed that the metal is atomically monodispersed in Pt=N₂=Fe ABA sample with less than 2.1% cluster, and

the monoatomic dispersed metal is dominated by dual-site structure (72%). Further regional intensity analysis (Fig. N24-25) showed that the dual-site structure was composed of Pt and Fe, which is consistent with our preparation strategy. Moreover, the interatomic distance in the dual-atom site structure obtained from the regional intensity analysis is about 2.85 Å, which is obviously larger than the known metal bond length in Pt foil (Pt-Pt, 2.76 Å), Fe foil (Fe-Fe, 2.42 Å) and PtFe nanoparticles (Pt-Fe, 2.55 Å) [*Nat. Comm.* **2014**, 5, 5634; *Nat. Comm.* **2021**, 12, 2870]. Therefore, we judged that the local structure of metal sites is heteronuclear metal coordination. The strategy of FT-XAFS spectral fitting is based on the basic understanding of the metal local structure in the sample as analyzed above, and the Pt-Fe and Fe-Pt path are selected for the Pt L_3 -edge and Fe K -edge FT-XAFS spectra, respectively. The fitting result shows that the bond length of the metal coordination is about 2.86 Å (Table N5), which is consistent with the regional intensity analysis results in STEM-HAADF images. However, if the fitting path is changed to Pt-Pt or Fe-Fe, it seems unreasonable and lack of theoretical basis.

The actual atomic spacing of ~ 2.86 Å is a reasonable bond length based on the Pt=N₂=Fe configuration (Fig. N27), which can be supported by the fitting information that the coordination numbers (CNs) between metals and N atoms is 4, the coordination number between Pt and Fe is about 1, and the metal bond length is larger than that of Pt-Fe directly bonded.

The in-situ XAFS measurements were performed with catalyst-coated carbon cloth in O₂-saturated alkaline solution by a smart homemade in-situ cell. The catalyst-coated carbon cloth is closely attached to the surface of the in-situ cell on the side of incident light, which reduces the absorption of fluorescence signal by the solution and ensures more active sites to participate in the reaction, thereby improving the accuracy of measurement. The data under each potential is collected after the voltage is applied for 10 min to ensure the stability of the sample state. During the collection process, the data jump which represents the stability of the test device is paid attention to be in the normal value, so as to exclude the disturbance in the reaction cell and the interference of unstable light-source signal, and guarantee that the collected signal reflects the local

structure changes of the sample. At the same time, two independent measurements were conducted for each group of data, and the change trend remained consistent.

The monoatomic dispersion state of metal in the Pt AMS and Fe AMS samples can be demonstrated by the absence of metal diffraction peaks in XRD and the absence of metal coordination peak (located at 2-3 Å) in Pt L_3 -edge (Fe K -edge) FT-XAFS spectra of Pt AMS (Fe AMS) sample.

We realize that the expression of fitting path selection is not clear enough, so in the “Strategies of XAFS curve Fittings” part at the end of the revised manuscript, the following text **has been added**: “For the ex situ Pt=N₂=Fe ABA sample, the coordination peaks at 1.63 Å assigned to the Pt-N coordination with N atoms that was derived from the N-rich –NH₂ functionalized support. And based on the analysis of the dominant heteronuclear dual-site atomic spacing in Cs-corrected STEM-HAADF images and the location of metal coordination peak, the peak at 2.65 Å in Fourier-transformed curves should be assigned to Pt-N-Fe coordination. Therefore, two separate Pt–N and Pt-N-Fe scattering paths were included for fitting.”

Fig. N27 Schematic diagram of Pt=N₂=Fe configuration.

Table N5 Structural parameters of the Pt L_3 -edge and Fe K -edge for Pt=N₂=Fe ABA catalyst extracted from quantitative EXAFS curve-fitting using the ARTEMIS module of IFEFFIT.

Sample	Path	CNs	$R(\text{Å})$	$\sigma^2 (10^{-3} \text{Å}^2)$	ΔE_0 (eV)	R -factor
Pt L_3 -edge ex situ	Pt-N	4.1±0.2	2.01±0.02	5.2±1.3	7.5	0.005
	Pt-Fe	1.1±0.2	2.86±0.02	7.6±1.5		
Fe K -edge ex situ	Fe-N	4.2±0.2	1.95±0.02	6.3±1.0	6.3	0.004
	Fe-Pt	1.0±0.2	2.85±0.02	8.1±1.5		

3. Question: Figure 2A: Almost -7 mA/cm² for a MT limited current density is higher than expected. Can you explain this?

Reply: Thank you for your careful inspection. We have reconsidered this issue seriously. First of all, we performed normalized linear sweep voltammetry measurements for the synthesized samples. As shown in Fig. N28, the limiting current density of Pt=N₂=Fe ABA samples was 6.5±0.05 mA cm⁻² after subtracting the background current. Although it was slightly higher than the theoretical value of 6.16 mA cm⁻², it is within 10% of the experimental error [*J. Electrochem. Soc.* **2018**, 165, J3001-J3007; *Anal. Chem.* **2010**, 82, 6321]. In addition to the factors such as catalyst loading and electrolyte conditions, the slightly larger limit current density of the Pt=N₂=Fe ABA may be also attributed to the good mass transfer of the noble-metal catalyst [*ChemElectroChem* **2020**, 7, 1107–1114].

Fig. N28 ORR polarization plots of Pt=N₂=Fe ABA and reference samples.

4. Question: For performance normalization it is standard to compare surface areas of the materials tested. How do the electrochemical, BET, or other surface areas compare? Is it possible that the Pt=N₂+Fe ABA just has the highest number of available active sites? It is also standard with Pt based materials to normalize based on the mg of Pt. How do these materials compare to each other and other PGMs on a mA/mgPt basis?

Also, the TOF calculation was not clear. For the Pt=N₂=Fe is each pair considered an active site? How does the TOF change if each metal is considered?

Reply: Thank you for your deep consideration. We are sorry that the electrochemical intrinsic activity parameters were not clearly addressed in the previous manuscript. Electrochemical surface areas (ECSA) of the catalysts will be provided in the revised manuscript. And the normalized comparisons of mass activity (based on the Pt loading) and turnover frequency (TOF) will be further discussed in details in the revised manuscript.

For the synthesized platinum-based single-atom catalysts, we measured the electrochemically active surface areas (ECSA) by the double-layer capacitance (C_{dl}) based on cyclic voltammetry [*J. Am. Chem. Soc.* **2013**, 135, 16977–16987]. The cyclic voltammetry curves of Pt=N₂=Fe ABA catalyst and the reference samples with the scanning rates from 10 to 50 mV s⁻¹ were shown in Figure N29. The non-Faradaic capacitances associated with the electrochemically accessible surface area (ECSA) of the catalysts are reflected by current densities in the cyclic voltammograms. The gravimetric double layer capacitance C (F/g) can be related to the ratio of the capacitance current density J (mA cm⁻¹) to the scan rate range Δv (mV s⁻¹) and mass of catalyst deposited on the electrode (m , 1 g m⁻²): $C = J/\Delta v \cdot m$ [*Electrochim. Acta* **2004**, 49, 257-262]. Then the ECSA (m² g⁻¹) can be estimated as the specific value from the gravimetric capacitance C by the equation: $ECSA = C/C_s$, where C_s is the double layer capacitance (F m⁻²) of the glassy carbon electrode surface, for which the typical value of 0.4 F m⁻² was used in KOH solution [*ACS Catal.* **2022**, 12, 7994–8006]. The error bars were determined from the standard deviation of three individual measurements. As a result, the Pt=N₂=Fe ABA catalyst has the highest ECSA of 210.5 m² g⁻¹, corresponding to the highest ORR activity.

Considering that the mass activity (MA) and turnover frequency (TOF) are closely related to the metal loading, we carefully determined the mass content (wt%) of the metal in the samples by means of ICP-OES and XPS. In order to exclude the contribution from the superposition of the active site numbers, we controlled the metal loading to ensure that the three as-synthesized catalysts (Pt=N₂=Fe ABA, Pt AMS and

Fe AMS) contained approximately equal densities of metal molar number ($\sim 3.5 \times 10^{-7}$ mol mg^{-1}). According to the ICP results, the number of metal moles in the samples can be calculated: Pt 1.53×10^{-7} mol mg^{-1} and Fe 2.1×10^{-7} mol mg^{-1} in Pt=N₂=Fe ABA, Pt 3.0×10^{-7} mol mg^{-1} in Pt AMS and Fe 3.61×10^{-7} mol mg^{-1} in Fe AMS. Normatively, we calculated the mass activity of the prepared Pt=N₂=Fe ABA catalyst was 14.1 A $\text{mg}_{\text{Pt}}^{-1}$ at 0.90 V vs. RHE (Fig. N30) with the Pt loading of 2.9 $\mu\text{g}_{\text{Pt}} \text{cm}^{-2}$ on rotating disk electrode (RDE) according to the standardized ORR mass activity calculation formula of $\text{MA} = J_k / M_{\text{Pt}}$ (the kinetic current densities J_k , mA cm^{-2} ; the Pt loading density M_{Pt} , $\mu\text{g}_{\text{Pt}} \text{cm}^{-2}$), we calculated that the mass activity of the prepared Pt=N₂=Fe ABA catalyst was 14.1 A $\text{mg}_{\text{Pt}}^{-1}$ at 0.90 V vs. RHE. The mass activity achieved in the Pt=N₂=Fe ABA exceeds that of the best platinum-based catalysts reported previously (12.36 A $\text{mg}_{\text{Pt}}^{-1}$) [Science, **2018**, 362, 1276–128] and nearly doubles the outstanding reported mass activity value of $\sim 4\text{--}7$ A $\text{mg}_{\text{Pt}}^{-1}$ (Table N6). The mass activity of the reference catalysts (Pt/C, Pt AMS and Fe AMS) was calculated by the same formula based on the metal loading on electrode of 20 $\mu\text{g}_{\text{Pt}} \text{cm}^{-2}$ in commercial Pt/C, 5.6 $\mu\text{g}_{\text{Pt}} \text{cm}^{-2}$ in Pt AMS and 1.96 $\mu\text{g}_{\text{Fe}} \text{cm}^{-2}$ in Fe AMS. For the normalized comparison of turnover frequency (TOF), we recalculated it on the basis of platinum loading using the following formula:

$$\text{TOF} = \frac{J_k}{\frac{W_{\text{metal}}}{M_{\text{metal}}} \times N_A \times m_{\text{loading}}}$$

where W_{metal} and M_{Pmetal} are the mass fraction (wt%) and molar mass of Pt (195 g mol^{-1}). m_{loading} is the loading mass of the catalyst on RDE (g cm^{-1}). N_A is Avogadro's number (6.02×10^{23} mol^{-1}). J_k is the kinetic current density (mA cm^{-2}). Thus, the product of $\frac{W_{\text{metal}}}{M_{\text{metal}}} \times m_{\text{loading}}$ is the mole numbers of the active site in the loading catalyst on RDE [ACS Catal. **2020**, 10, 2452–2458; Angew. Chem. Int. Ed. **2021**, 60, 4049–4054]. The TOF of Pt sites in Pt=N₂=Fe ABA, commercial Pt/C and Pt AMS are 15.3 $\text{e}^- \text{s}^{-1} \text{site}^{-1}$, 0.48 $\text{e}^- \text{s}^{-1} \text{site}^{-1}$ and 0.37 $\text{e}^- \text{s}^{-1} \text{site}^{-1}$ at 0.90 V vs. RHE, respectively. It is worth noting that TOF reflects the conversion efficiency of the catalytic active site in the catalyst, thus the Pt=N₂=Fe pair should be regarded as a unit of synergistic active site in the Pt=N₂=Fe ABA catalyst due to the actual dual-site catalytic mechanism. And for

the sake of computational rigor, when calculating the numbers active site of Pt=N₂=Fe ABA catalyst, we included Pt=N₂=Fe dual-site pair and single active sites in a statistical proportion (about 7:3). Although the TOF value of Pt=N₂=Fe ABA is still higher than that of the reference samples (Pt AMS and Fe AMS) even if each metal site in Pt=N₂=Fe ABA is included in the calculation (as the total numbers of metal mole is roughly equal, $\sim 3.5 \times 10^{-7}$ mol mg⁻¹), this calculation method of active sites does not seem to match the actual catalytic reaction mechanism.

Accordingly, Figure N30 **has been added** in the revised manuscript as Fig. 2d. Figure N29 and Table N6 have been added in Supplementary Information as Supplementary Fig. 16 and Supplementary Table 7. And in line 12, page 9 of the revised manuscript, the following text **has been added**: “*Pt=N₂=Fe ABA displays an ultrahigh TOF of 15.3 e⁻·site⁻¹·s⁻¹ and a great MA of 14.1 A mg_{Pt}⁻¹ at 0.90 V vs. RHE, 32 and 61 times higher than that of commercial Pt/C reference (0.48 e⁻·site⁻¹·s⁻¹, 0.23 A mg_{Pt}⁻¹), and the high MA of platinum in Pt=N₂=Fe ABA exceeds most of the reported Platinum-based catalyst (Supplementary Table 7). The cyclic voltammetry curves of Pt=N₂=Fe ABA catalyst and the reference samples with the scanning rates from 10 to 50 mV s⁻¹ are shown in Supplementary Fig. 16. As a result, the Pt=N₂=Fe ABA catalyst has the highest electrochemically active surface areas (ECSA) of 210.5 m² g⁻¹, corresponding to the highest ORR activity.*”

Fig. N29 a-c The cyclic voltammetry curves of Pt=N₂=Fe ABA catalyst and the reference samples with the scanning rates from 10 to 50 mV s⁻¹. **d** ECSA values calculated from potential cycling.

Fig. N30 The turnover frequency and mass activity of the Pt=N₂=Fe ABA and reference samples at 0.9 V vs. RHE. (The standard of mass activity for Fe AMS sample is A mg_{Fe}⁻¹.)

Table N6 Comparison of RDE mass activity of Pt=N₂=Fe ABA with several high-performance catalysts from published works.

Table N6 Comparison of RDE mass activity of Pt=N₂=Fe ABA with several high-performance catalysts from published works.

Catalyst	Mass activity A·mg ⁻¹ _{Pt} @0.9 V vs. RHE	Reference
Pt=N ₂ =Fe ABA	14.1	This work
LP@PF-2	12.36	Science 2018, 362, 1276–1281
Mo-Pt ₃ Ni/C	6.98	Science 2015, 348, 1230–1234
Pt ₃ Ni/C nanoframe	5.7	Science 2014, 343, 1339–1343
PtNiCo NW	4.2	Sci. Adv. 2017, 3, 1601705
sd-Pt ₈₄ Ni ₁₂ Co ₄	7.1	Nat. Catal. 2022, 5, 513–523
PtCo-NC-3	4.16	Research 2020, 9167829
J-PtNWs/C	13.6	Science 2016, 354, 1414–1419

5. Question: Figure 4: Fe and Pt are known to have different ORR mechanisms. Because there are so few references in the literature for this *O-O* species it seems vital to have an Fe AMS control to make sure that the signal attributed to the O-O is not something that would have been seen for an Fe-only active site. Specifically, after looking at the literature it seems as though a “free superoxide species (O₂⁻), typically absorbs in the range 1,200–1,070 cm⁻¹ (10.1038/nchem.1874)”

Reply: Thank you for your constructive comments. We have added the in-situ FTIR measurement results of Fe AMS catalyst (Fig. N31), and further discussed the catalytic mechanism reflected by the in-situ FTIR characterization. In Fig. N30, the newly formed infrared vibration peak located at 980 cm⁻¹ is similar to the Pt AMS sample, and the vibration can be assigned to the emergence of *OOH intermediate species. As the reviewer mentioned, different active metals, such as typical Pt and Fe, undergo inconsistent rates of catalytic steps due to different adsorption Gibbs free energy of key intermediates (such as *OOH, *O, and *OH), but the rate-determining step of ORR on single-atom dispersed Pt or Fe site are both the conversion of O₂ into *OOH [*ACS Catal.* **2021**, 11, 6304–6315; *Chem* **2019**, 5, 2099–2110]. Thus, the rate-determining step that determines the key intermediate *OOH was observed in the in-situ FTIR

characterization of both Pt AMS and Fe AMS. In contrast, vibration peak with obviously different positions at 1115 cm^{-1} was observed in the in-situ FTIR of Pt=N₂=Fe ABA, which can be attributed to the *O-O* radicals bound to bimetals at around 1100 cm^{-1} [*Energy Environ. Sci.* **2020**, 13, 2200-2208; *Chem* **2021**, 7, 2101-2117]. By comparing the results of forming different intermediates for Pt=N₂=Fe ABA and the reference samples (Pt AMS and Fe AMS), it can be concluded that synergistic dual-site in Pt=N₂=Fe ABA can promote the activation of O₂ to form *O₂. And the absorption peak intensity of the *O-O* intermediate decreases rapidly after the in-situ potential of 0.90 V vs. RHE was applied, which implies that the Pt-O-O-Fe structure promotes the rapid and direct cleavage of *O₂ radical.

In addition, the in-situ SR-FTIR specializes in the detection of easily accumulated reaction intermediates in the rate-limiting step. The activation of O₂ to superoxide species generally occurs at metal sites, followed by further M-OOH formation on single metal site or M-O-O-M cleavage at metal dual-site. And the superoxide species (O²⁻) in free is difficult to capture.

Accordingly, Figure N31 **has been added** in Supplementary Information as Supplementary Fig. 25. In line 14, page 13 of the revised manuscript, the following text **has been added**: “*Meanwhile, the in-situ SR-FTIR measurement was also conducted for the Fe AMS sample to make a good comparison (Supplementary Fig. 25). The newly formed infrared vibration peak located at 980 cm^{-1} , which is similar to the Pt AMS sample, informing the formation of *OOH on the single iron site within Fe AMS. It can be seen that *O-O* radical with obviously different peak position (1115 cm^{-1}) only appears for Pt=N₂=Fe ABA, suggesting that O₂ is effectively activated into *O₂ and cleaves directly.*”

Fig. N31 600–1500 cm^{-1} range of in-situ SR-FTIR characterization for Fe AMS.

6. Question: Error bars from the RDE samples tested in triplicate (main text), as well as nitrogen CVs and the forward and backward sweep (SI) should be included to demonstrate that this is a reproducible performance and clarify the materials entire electrochemical profile.

Reply: Thank you for your suggestion that is quite useful for improving the quality of this work. We conducted three individual measurements of the RDE-based performance curves of each catalyst in the manuscript (Figure 2), and the error intervals is marked on the curves with error bars.

The cyclic voltammetry in N_2 -saturated 0.1 M KOH electrolyte was employed to verify the electrochemical properties of metals in the as-synthesized platinum-based catalysts. As can be seen from the cyclic voltammetry curves of Pt= N_2 =Fe ABA and Pt AMS (Fig. N32), no characteristic peaks of hydrogen adsorption and/or desorption were observed due to the atomic-scale platinum dispersion in the catalyst.

Long-term durability tests were conducted by holding at a constant potential of 0.7 V vs.RHE for 100 hours during the ORR. The polarization plots and the cyclic voltammetry curves before and after constant potential tests are shown in Fig. N33. After the 100 hours test at 0.7 V vs. RHE, the half-wave potential only showed a drop of 15 mV and the capacitance increase (inset of Fig. N33) is less than 4%, suggesting that the active site has excellent stability and carbon oxidation resistance.

Accordingly, Figure N32 and 33 **have been added** in Supplementary Information as Supplementary Fig. 15 and 18. and in line 3, page 8 of the revised manuscript, the following text **has been added**: “The cyclic voltammograms of Pt=N₂=Fe ABA and Pt AMS are recorded in N₂-saturated 0.1 M KOH electrolyte, which present no the hydrogen adsorption and/or desorption peaks characteristic of Pt due to the atomic-scale metal dispersion (Supplementary Fig. 10).” And in line 4, page 10, the following text **has been added**: “The Pt=N₂=Fe ABA catalyst demonstrates reliable stability with better carbon oxidation and methanol resistance in chronoamperometry measurements, in which the E_{1/2} only drops 15 mV and the capacitance increases less than 4% after continuous operation for 100 h (Supplementary Fig. 17-19).”

Fig. N32 Cyclic-voltammogram of Pt=N₂=Fe ABA and Pt AMS catalysts recorded at the potential range of 0–1.2 V vs. RHE and 50 mV s⁻¹ scan rate in an N₂-saturated 0.1 M KOH electrolyte.

Fig. N33 Steady-state ORR polarization plots and cyclic voltammetry curves before and after constant potential tests at 0.7 V for 100 hours in O₂-saturated 0.1 M KOH electrolyte.

7. Question: Random: AMS is not defined until after it shows up in Figure 1. If this material is breaking Pt scaling relations it seems like it should be more active than all Pt.

Reply: We are grateful to the reviewer for your careful inspection. To make the atomic-scale monometallic site samples (Pt AMS and Fe AMS) more clear, we have supplemented the definition of the Pt AMS and Fe AMS catalysts in the experimental part of the manuscript.

In reverse line 15, page 5 of the revised manuscript, the following text **has been added:** *“The atomically monometallic site catalysts (Pt AMS and Fe AMS) are synthesized for reference in similar process, except that the precursor solution is replaced with H_2PtCl_6 or $FeCl_3 \cdot 6H_2O$ aqueous solution, respectively.”*

8. Question: Please include a table of XPS fits and details from fits in the SI (e.g. peak positions, ratios, etc.)

Reply: Thank you for your constructive suggestion that is quite useful for improving the quality of this work. The high-resolution N1s, Pt 4f and Fe 2p XPS spectra of Pt=N₂=Fe ABA sample were fitted in this work to obtain the structure and valence state information. In order to make the fitting results more clear, we supplemented the table of fitting details according to the suggestion of reviewers.

For the N 1s XPS spectra, the fitting results display four peaks of pyridinic N (398.26 eV), graphitic N (400.78 eV), oxidized N (403.01 eV), and N coordinated with metal sites (denoted as Pt/Fe–N) (399.35 eV) (Fig. N34) [*Adv. Mater.* **2020**, 32, 2003577], more details found in Table N7. For the Pt 4f XPS spectra, the fitting results display two peaks of Pt 4f_{5/2} (75.85 eV) and Pt 4f_{7/2} (72.54 eV) (Fig. N35a, Table N8). The difference value of the two binding energy is 3.31 eV and the area ratio of 4f_{5/2} and 4f_{7/2} is 1.33, which meet the fitting criterion of Pt 4f split energy level [CAS Registry No:7440-06-4; *Arkiv Fur Fysik* 1968, 38, 349]. The Fe 2p XPS spectra was fitted to two peaks of Fe 2p_{3/2} at 710.36 eV and Fe 2p_{1/2} at 723.40 eV, respectively (Fig. N35a, Table N9). The difference value of the two binding energy is 13.04 eV with the area

ratio of 1.6, which meet the fitting criterion of Fe 2p split energy level [CAS Registry No:7439-89-6; *Phys. Scripta* 23, 825 (1981)].

Accordingly, Fig. N34 and N35 have been added in the Supplementary Information as Supplementary Fig. 10-11. Table N7, N8 and N9 have been added in the Supplementary Information as Supplementary Table 1-3. In line 17, page 6 of the revised manuscript, the following text **has been added**: “the fitting results of Pt 4f and Fe 2p XPS spectra for Pt=N₂=Fe ABA show that the metal atoms have a positive charge state with no zero-valence species appearing (Supplementary Fig. 9, Supplementary Table 1-3).”

Fig. N34 The fitted N1s XPS spectra of Pt=N₂=Fe ABA sample.

Fig. N35 The fitted Pt 4f and Fe 2p XPS spectra of Pt=N₂=Fe ABA sample.

Table N7 Fitting parameters of N 1s XPS spectra for Pt=N₂=Fe ABA sample.

Fitting of N 1s	Position (eV)	FWHM (eV)	Area
Pyridinic N	398.26	1.32	14640.93
Pt/Fe-N	399.35	2.58	15931.34
Graphitic N	400.78	1.82	20752.82
Oxidized N	403.01	3.60	4900.56

Table N8 Fitting parameters of Pt 4f XPS spectra for Pt=N₂=Fe ABA sample.

Fitting of Pt 4f	Position (eV)	FWHM (eV)	Area
Pt 4f _{5/2}	75.85	1.25	11748.9
Pt 4f _{7/2}	72.54	1.27	8823.47

4 : 3

Table N9 Fitting parameters of Pt 4f XPS spectra for Pt=N₂=Fe ABA sample.

Fitting of Fe 2p	Position (eV)	FWHM (eV)	Area
Fe 2p _{3/2}	710.36	3.5	12045.29
Fe 2p _{1/2}	723.40	3.5	7601.47
Fe 2p _{3/2} sat.	715.10	4.2	5806.03
Fe 2p _{1/2} sat.	728.56	4.1	3504.41

1.6 : 1

9. Question: Figure 1C: Based on the HAADF-STEM, what is the range in distance between the Fe and Pt?

Reply: Thank you for your nice question. we distinguish the two metals (Pt and Fe) based on the commonly adopted theory that the contrast intensity of atoms in the Cs-corrected STEM-HAADF images highly depends on the atomic number (Z) [*Nat. Nanotechnol.* **2022**, 17, 590-602]. For the Cs-corrected STEM-HAADF images with high-magnification acquired from the Pt=N₂=Fe ABA sample (Fig. N24-26, as stated in **reply to question #1 Figure 1A-N**), the dominant dual-site structure can be clearly observed and accounts for the majority. Further, intensity analysis was performed at four dual-atom-like regions of STEM-HAADF images (Fig. N24d-e and Fig. N25b). All the two bright spots along the arrow show different intensity, confirming that the dual-site is composed of Pt and Fe atoms. In the meanwhile, the atomic spacing in the dual-site structure can be obtained from the intensity profile that are about 0.83-0.91 Å,

which is corresponding to the fitting results of metal coordination bond length (0.86 Å) in FT-EXAFS spectra.

In addition, in line 1, page 6 of the revised manuscript, the following text **has been added**: *“Based on the fact that the intensity of atoms in the STEM-HAADF image highly depends on their atomic number (Z), intensity analysis was performed at four dual-site regions in STEM-HAADF images to clarify the composition of the dual-site structure (Fig. 1c and Supplementary Fig. 8d-e). All the two bright spots along the arrow show different intensity, confirming that the dual-site is composed of Pt and Fe atoms. The statistical analysis results also show that the atomic distance between Pt and Fe is approximately 2.81–2.93 Å.”*

10. Question: The Pt/C control tested seems less active and stable than standard commercial Pt/C. How does this compare to literature values of commercial Pt?

Reply: Thank you for your nice question. The ORR performance tests for commercial Pt/C catalyst will be analyzed and compared in detail. In this work, the commercial 20 wt% Pt/C was used to perform performance tests with the Pt loading of $20 \mu\text{g}_{\text{Pt}} \text{cm}^{-2}$ on glassy carbon working electrode. At least three independent measurements are taken for each test to obtain reliable data within the error range. As a result, the half-wave potential, an important evaluation index of ORR catalyst, is 0.86 V vs. RHE, and the current decayed to 65.38 % after 60 hours of chronoamperometry test at 0.7 V vs. RHE. Based on a study of the meta-analysis of commercial Pt/C measurements [*Chin. J. Catal.* **2022**, 43, 116–121], the half-wave potential of 0.84 ± 0.03 V vs. RHE with $20 \mu\text{g}_{\text{Pt}} \text{cm}^{-2}$ Pt loading of commercial 20 wt% Pt/C is suggested as the “Golden reference” for the performance evaluation of other ORR catalysts. The activity index of the standard commercial Pt/C obtained in our work seems to meet the “Golden reference”. In addition, the comparison of activity and stability values with those in some typical literature are listed in Table N10.

According to the comparison results in the table, the test values of commercial Pt/C activity and stability can be used as a reference for other catalysts in our work.

Table N10. Comparison of ORR performance in 0.1 M KOH of commercial Pt/C in the literatures.

Pt/C catalysts	Pt loading ($\mu\text{g}_{\text{Pt}} \text{cm}^{-2}$)	$E_{1/2}$ (V vs RHE)	J (mA cm^{-2})	Stability (Norm. Current,%)	Reference
20 wt%	20	0.851	5.91	65.38 % after 60 h	This work
20 wt%	60	0.868	5.55	66.1 % after 50 h	Adv. Mater. 2021, 33, 2104718
20 wt%	24.2	0.86	5.20	79 % after 5.5 h	Adv. Sci. 2021, 8, 2101344
20 wt%	20	0.85	5.51	78 % after 10 h	Angew. Chem. Int. Ed. 2021, 60, e20211444
20 wt%	20	0.86	5.75	42% after 10 h	ACS Catal. 2021, 11, 6304
20 wt%	20.4	0.840	6.2	-	Nat. Commun. 2018, 9, 5422

11. Question: Does the oxidation state fits from XPS and XAS match for the Pt and Fe? Please include quantitative analysis comparison.

Reply: Thank you for your nice question. The oxidation state of Pt and Fe in the Pt=N₂=Fe ABA sample were analyzed in detail by X-ray photoelectron spectroscopy (XPS) and X-ray absorption structure (XAS) spectra characterization.

The valence states of the metal species can be described quantitatively from the X-ray absorption near-edge structure (XANES) spectra. Fig. N36a shows the Pt L₃-edge XANES spectra of the Pt=N₂=Fe ABA sample and the reference samples (Pt foil and PtO₂). Due to the Pt L₃-edge XANES spectra corresponding to the transition from the occupied Pt 2p_{3/2} core-electron to empty 5d states, the valence state is determined by the intensity of the white line peak [*Nat. Commun.* **2019**, 10, 440; *Nat. Commun.* **2020**, 11, 1029]. The normalized XANES spectra by subtracting the spectra of Pt=N₂=Fe ABA and PtO₂ sample to that of Pt foil reference is shown in Fig. N36b. The oxidation states are fitted through integrating the area of the white-line peak in Fig. N36b from 11562.0 to 11580.0 eV. According to the area integral results and the known valence state (+4) of PtO₂, the valence state of Pt in the Pt=N₂=Fe ABA sample can be calculated as +3.36 (Fig. N36c). In the XPS characterization, as shown in Fig. N35a, the Pt 4f XPS spectrum can be fitted into two peaks of Pt 4f_{5/2} (75.85 eV) and Pt 4f_{7/2} (72.54 eV). Referring to the standard binding energies of Pt²⁺ and Pt⁴⁺ (dotted line in Fig. N), the oxidation state of Pt in Pt=N₂=Fe ABA is between +2 and +4 and slightly

close to +4, which is consistent with the results of XANES quantitative analysis [Science **2015**, 350, 189; J. Am. Chem. Soc. **2015**, 137, 3470].

The Fe 2p XPS spectrum is fitted to a peak of Fe 2p_{3/2} at 710.36 eV and a peak of Fe 2p_{1/2} at 723.40 eV. Considering that the binding energy are near that of the 2p orbital for the Fe²⁺ and Fe³⁺ species, and far from the binding energy of metallic Fe⁰ typically at 710.36 eV (2p_{3/2}) and 721.8 eV (2p_{1/2}), the Fe specie in the Pt=N₂=Fe ABA sample is in the positive oxidation states between +2 and +3. According to the Fe K-edge XANES spectra (Fig. N36a), the energy absorption edge of the Pt=N₂=Fe ABA sample between that of Fe foil and Fe₂O₃, which represents the 1s→3d transition, revealing an Fe^{δ+} oxidation state in Pt=N₂=Fe ABA. The oxidation state can be quantitatively fitted on the basis of the difference value (ΔE_0) of the absorption edge [Nat. Cat. **2019**, 2, 134]. As can be seen from the Fig. N36b, the fitting results of Fe specie in Pt=N₂=Fe ABA is +2.40, which verifies the results of valence state analysis in XPS characterization.

Accordingly, Figure N36 and N37 **have been added** in Supplementary Information as Supplementary Fig. 12-13, and in reverse line 14, page 6 of the revised manuscript, the following text **has been added**: “*the fitting results of Pt 4f and Fe 2p XPS spectra for Pt=N₂=Fe ABA show that the metal atoms have a positive charge state with no zero-valence species appearing (Supplementary Fig. 10-11, Supplementary Table 1-3). The oxidation state of Pt and Fe were further analyzed quantitatively from the X-ray absorption near-edge structure (XANES) spectra. The normalized Pt L₃-edge XANES spectra by subtracting the spectra of Pt=N₂=Fe ABA and PtO₂ sample to that of Pt foil reference are shown in Supplementary Fig. 12b. The valence state of Pt in the Pt=N₂=Fe ABA sample can be calculated as +3.36 (Supplementary Fig. 12c) through the area integral of normalized white-line peak, which is consistent with the valence state analysis in XPS characterization. The oxidation state of Fe was quantitatively evaluated as +2.40 based on the difference value (ΔE_0) of the absorption edge, verifying the XPS results. (Supplementary Fig. 13)*”

Fig. N36 (a) Pt L_3 -edge XANES spectra for Pt=N₂=Fe ABA sample and the reference standards of Pt foil and PtO₂. (b) Normalized difference spectra for Pt L_3 -edge XANES using Pt foil as reference. The oxidation states are fitted through integrating the area of the white-line peak from 11560.0 to 11580.0 eV. (c) The fitted oxidation states of Pt from Δ XANES spectra.

Fig. N37 (a) Fe K -edge XANES spectra for Pt=N₂=Fe ABA sample and the reference standards of Fe foil and Fe₂O₃. (b) The fitted oxidation states of Fe from XANES spectra.

REVIEWER COMMENTS

Reviewer #1 (Remarks to the Author):

The authors addressed a number of issues raised in the initial review and the revision read much better. Yet critical issues remain, as detailed below.

1. The authors claimed that the sample contained 72.6% dual sites, 25.3% SA sites and 2.1% clusters (Fig S7). Yet in the XAS fitting, only the dual site structure was used. This raises questions about the validity of the fitting results.

2. XPS and ICP results showed that the sample actually contained about 60% more Fe than Pt (Fe:Pt = 1.6). The impacts of this excess of Fe need to be taken into account as Fe atomic sites are active in alkaline media towards ORR.

3. The O 1s spectra need to be included in the SI to evaluate the formation of M-O species.

4. The authors appeared to overload the catalysts onto the electrode surface, as evidenced in Fig S20a which showed a peak around +0.8 V (red curve). Too thick a layer will allow self-disproportionation reaction of any peroxide species produced, leading to overestimation of n . This error has appeared in a number of reports in the literature.

5. How did the authors derive the Tafel plots in Fig 2c? The range of potential is inconsistent with the data in Fig 2a. Also, in Fig 2e, for the diffusion-controlled region, you do not use the K-L equation; you use the Levich equation instead. There is no point of plotting the data at various potential positions within the diffusion-controlled region.

6. The ZAB performance, while good, is not exceptional as compared to the state of the art. For instance, in a study by Liao et al (ACS Appl. Mater. Interfaces 2019, 11, 47, 44153–44160), they reported an ultrahigh power density of 479.1 mW cm⁻², more than twice that reported here. Therefore, the question of whether the scaling relation was broken remains unanswered, as one would anticipated a much enhanced performance.

7. The authors need to discuss/mention all figures (in particular those in the SI) in proper order.

Reviewer #2 (Remarks to the Author):

The authors have made substantial efforts to address the points raised by this reviewer and other reviewers. The revised manuscript appears to be significantly improved compared to the previous version. This reviewer recommend the publication of this paper.

Reviewer #3 (Remarks to the Author):

The authors did a thorough re-evaluation of their paper. Especially with the softening of their claims of confinement I believe that they have adequately addressed all of my previous concerns.

Response to Reviewers' Comments

We are grateful to the reviewers for having given us valuable comments on our manuscript NCOMMS-22-14848A-Z. We have read the reviewers' comments seriously. The detailed replies to the questions are presented in a point-to-point manner as follows.

To Reviewer #1:

General comment: The authors addressed a number of issues raised in the initial review and the revision read much better. Yet critical issues remain, as detailed below.

Author Reply: We are very grateful to the reviewer's encouraging comments, which greatly improve the quality of our manuscript. According to your nice suggestions, we have made corresponding modifications on our manuscript as listed below.

1. Question: The authors claimed that the sample contained 72.6% dual sites, 25.3% SA sites and 2.1% clusters (Fig S7). Yet in the XAS fitting, only the dual site structure was used. This raises questions about the validity of the fitting results.

Author Reply: Thank you for your comprehensive consideration. The discussions about the XAS fitting are stated as follows.

The atomically monometallic sites (25.3%) in the Pt=N₂=Fe ABA sample only have the first-adjacent non-metallic coordination (M-N_x), and its coordination environments are consistent with the first-adjacent non-metallic coordination of the metals in dual-sites (PtN_x-FeN_x), and the two are indistinguishable in the EXAFS results during 1–2 Å, so that it is difficult to fit them separately in the actual EXAFS fitting operation. Therefore, the fitting results of the sample in the first-adjacent coordination (located at 1–2 Å) actually already contain the information of both the monometallic sites and dual-sites (97.9% in total), where the cluster (2.1%) has no contribution to the first-adjacent non-metallic coordination.

The second-adjacent metal coordination peaks in Pt *L*₃-edge and Fe *K*-edge

FT-EXAFS spectra of Pt=N₂=Fe ABA were observed at 2.63 and 2.60 Å, respectively (Fig. N1), which are larger than the location of directly bonded Fe-Fe (2.15 Å), and Pt-Fe (2.25 Å) coordination peaks and close to the Pt-Pt (2.57 Å) coordination peak [*Nat. Comm.* **2014**, 5, 5634; *Nat. Catal.* **2022**, 5, 503-512], indicating that the contribution of the Fe or Pt-Fe clusters can be excluded. Thus, the second-adjacent metal coordination is mainly due to the contribution of the Pt-N-Fe coordination in the dual-sites and may contain a low proportion of Pt clusters contribution. Since the clusters (account for 2.1 %) in the Pt=N₂=Fe ABA sample consisting of a few atoms or atomic clusters are smaller than 1.5 nm, their nearest metal-metal coordination numbers are usually about 3~4 [*Nat. Catal.* **2022**, 5, 485–493], which means that the contribution of these clusters to the second-adjacent metal coordination of the Pt=N₂=Fe ABA sample is in a range of 0.08-0.12 (calculated by (3~4) * (2.1% / (72.6%+2.1%))). This contribution from the clusters is too small and can be ignored within the uncertainty (± 0.2) of the fitting for the second-adjacent metal coordination of samples.

Based on the above analysis, the structures of monometallic sites and dual-sites that account for 97.9 % have been included in the EXAFS fitting. In general, EXAFS fitting is based on the FEFF software by using the least square method, where the goodness of EXAFS fitting results is reflected by the most critical indicator, R-factor. Reliable fitting results require an R-factor < 2%. As the fitting results shown in Fig. N1c and Table N1, the R-factor is 0.004-0.005, much less than 0.02, suggesting the employed fitting model is reasonable. In addition, the peak positions in EXAFS are only a rough representation of atomic spacing because of the phase shift produced by single/multiple scattering. The fitted bond length of Pt-N-Fe, which can accurately reflect the coordinated atomic spacing, is about 2.86 Å based on the XAFS fitting (Table N1). Thus, the fitted bond length is consistent well with the atomic spacing analysis results in HAADF images (around 2.86 Å), proving that the fitting results are reasonable and reliable.

To make the presentation clear, in reverse line 12, page 7 of the revised manuscript,

the following text **has been added**: “To obtain the quantitative coordination information in the Pt=N₂=Fe ABA catalyst, FT-EXAFS curve-fitting analysis was performed based on the structures of dual-sites (72.6%) and monometallic sites (25.3 %).”

Fig. N1 a,b. Comparison of the FT-EXAFS spectra for Fe *K*-edge (**a**) and Pt *L*₃-edge (**b**) of Pt=N₂=Fe ABA sample. **c**, FT-EXAFS spectra of Pt *L*₃-edge and Fe *K*-edge, and the corresponding fitting curves for Pt=N₂=Fe ABA catalyst.

Table N1 Structural parameters of the Pt *L*₃-edge and Fe *K*-edge for Pt=N₂=Fe ABA catalyst extracted from quantitative EXAFS curve-fitting using the ARTEMIS module of IFEFFIT.

Sample	Path	CNs	R (Å)	σ^2 (10 ⁻³ Å ²)	ΔE_0 (eV)	R -factor
Pt L ₃ -edge ex situ	Pt-N	4.1±0.2	2.01±0.02	5.2±1.3	7.5	0.005
	Pt-N-Fe	1.1±0.2	2.86±0.02	7.6±1.5		
Fe K -edge ex situ	Fe-N	4.2±0.2	1.95±0.02	6.3±1.0	6.3	0.004
	Fe-N-Pt	1.0±0.2	2.85±0.02	8.1±1.5		

2. Question: XPS and ICP results showed that the sample actually contained about 60% more Fe than Pt (Fe:Pt = 1.6). The impacts of this excess of Fe need to be taken into account as Fe atomic sites are active in alkaline media towards ORR.

Author Reply: We are grateful to the reviewer for your careful inspection. Your suggestion that the excess Fe active sites should be taken into account is reasonable, and actually we have done so. The metal contents of the as-synthesized Pt=N₂=Fe ABA samples were confirmed by ICP and XPS measurements, and the Fe element

does exhibit a slightly higher content than Pt (Fe : Pt = 1.5 ± 0.1). For the calculation of the activity parameter (TOF) which reflects the conversion efficiency of the catalytic active site in the catalyst, both the Pt-Fe dual-sites and the excess single active sites were included. The turnover frequency (TOF) is calculated using the following formula:

$$\text{TOF} = \frac{J_k}{\frac{m_{\text{metal loading}}}{M_{\text{metal}}} \times N_A}$$

where $m_{\text{metal loading}}$ and M_{metal} are the metal loading mass on electrode (g cm^{-1}) and metal molar mass (g mol^{-1}). N_A is Avogadro's number ($6.02 \times 10^{23} \text{ mol}^{-1}$). J_k is the kinetic current density (mA cm^{-2}). It can be seen that the $\frac{m_{\text{metal loading}}}{M_{\text{metal}}}$ is the mole density of the metal active sites on the electrode (mol cm^{-1}). For counting the active sites number in Pt=N₂=Fe ABA catalyst, the Pt-Fe pair was regarded as a unit of synergistic active site due to the actual dual-sites catalytic mechanism. At the same time, the statistical proportion (3:1) of the dual-site pairs and single active sites was also taken into account. In this case, three quarters of the molar density ($\text{mol mg}_{\text{catalyst}}^{-1}$) of Pt in the catalyst was considered as the molar density value of dual-sites, and the rest was considered as the single Pt active sites. The fraction of the Fe molar density in catalyst minus the molar density value of dual-sites was included in the calculation as single Fe active sites. Thus, the molar density of the active sites in Pt=N₂=Fe ABA catalyst includes three parts: dual-sites, single Pt active sites and single Fe active sites. By the way, the calculation of the mass activity (MA) was based on the Pt loading mass ($\text{A mg}_{\text{Pt}}^{-1}$) and therefore does not involve the issues of Fe content.

Moreover, to exclude the contribution of the monometallic sites to the improved activities in Pt=N₂=Fe ABA sample, we also prepared the atomically monometallic sites samples (Pt AMS and Fe AMS) with the roughly equal metal molar contents to the Pt=N₂=Fe ABA for comparison ($\sim 3.5 \times 10^{-7} \text{ mol mg}_{\text{catalyst}}^{-1}$). Based on the electrochemical performance results, it can be found that the improved catalytic performance of the Pt=N₂=Fe ABA sample can be attributed to the Pt-Fe dual-sites,

rather than the monometallic sites.

Accordingly, in the “Methods” section of the revised manuscript, the following text **has been added**: “*The Pt=N₂=Fe pair was regarded as a unit of synergistic active site, and the dual-sites and single Pt/Fe active sites were taken into account in a statistical proportion (about 3:1).*” And in line 9, page 9 of the revised manuscript, the following text **has been added**: “*Based on the roughly equal numbers of active sites in the as-synthesized catalysts, the improved catalytic performance of the Pt=N₂=Fe ABA is mainly attributed to the formed Pt-N-Fe dual-sites.*”

3. Question: The O 1s spectra need to be included in the SI to evaluate the formation of M-O species.

Author Reply: Thank you for your constructive suggestion that is quite useful for improving the quality of this work. The O 1s XPS spectrum of Pt=N₂=Fe ABA has been supplemented to analyze the coordination environment of metals (Fig. N2). The unsmooth signal-to-noise ratio (black circle) was attributed to the low oxygen content in the Pt=N₂=Fe ABA sample. The fitted peak of O-H bond at 531.6 eV is associated with surface-adsorbed H₂O molecules [*Angew. Chem.* **2015**, 127, 7507–7512], and the fitted peak of O=N bond at 533.9 eV arises from the residual -NO₂ group that has not been completely replaced in the precursor [*J. Am. Chem. Soc.* **1984**, 106, 2758]. It is known that the characteristic peak for metal oxides or O-coordinated monoatomic metals (M-O) is usually located at 529-530 eV [*Adv. Energy Mater.* **2018**, 8, 1701347; *Sci. Adv.* **2020**, 6, eaba6586]. The O 1s XPS spectrum in Fig. N2 clearly shows the absence of the M-O species at around 529-530 eV.

Accordingly, Fig. N2 **has been added** in Supplementary Information as Supplementary Fig. 10b, and in line 12, page 6 of the revised manuscript, the following text **has been added**: “*The O 1s XPS spectrum in Supplementary Fig. 10b shows the absence of M-O bond, suggesting the inexistence of metal oxide species or O-coordinated monoatomic metals in Pt=N₂=Fe ABA sample.*” And in the caption of Supplementary Fig. 10, the following text **has been added**: “*The fitted peak of O-H*

bond at 531.6 eV is associated with surface-adsorbed H₂O molecules, and the fitted peak of O=N bond at 533.9 eV arises from the residual -NO₂ group that has not been completely replaced in the precursor.”

Fig. N2 High-resolution O 1s X-ray photoelectron spectroscopy (XPS) spectra of Pt=N₂=Fe ABA.

4. Question: The authors appeared to overload the catalysts onto the electrode surface, as evidenced in Fig S20a which showed a peak around +0.8 V (red curve). Too thick a layer will allow self-disproportionation reaction of any peroxide species produced, leading to overestimation of n. This error has appeared in a number of reports in the literature.

Author Reply: Thank you for your professional comments. We have tested the ORR performance of Pt=N₂=Fe ABA catalyst in acidic electrolyte with different loading amount on electrode surface. In Fig. N3a, when the loading of catalyst increased to 0.12 mg cm⁻², there was still an obvious disproportionation peak at around 0.8 V vs. RHE. And the peak disappeared as the loading decreased to 0.08 mg cm⁻². These results indicate that the disproportionation peak at around 0.8 V vs. RHE may be caused by the overloading of the Pt=N₂=Fe ABA catalyst, which has been mentioned by the reviewer. The disproportionation reaction that produces superoxide species at the original loading of 0.1 mg cm⁻² may be related to the H⁺ concentration and HO₂^{*} mass transfer around the electrode surface in the acidic electrolyte. [ACS Catal. 2020, 10, 852-863] Note that with the same catalyst loading of 0.1 mg cm⁻², the disproportionation peak was not observed in the polarization curves tested in alkaline

electrolyte, indicating that the loading amount is reasonable for alkaline electrolyte. Moreover, we also tested the H_2O_2 yield and electron transfer number (n) of the samples with different loading amount on RRDE. As shown in Fig. N3b, the electron transfer number at the loading of 0.08 mg cm^{-2} was the closest to 4.

Accordingly, the polarization curve tested with the catalyst loading of 0.08 mg cm^{-2} in Fig. N3a (dark curve) **has been added** in Supplementary Fig. 21a (red curve). And in the caption of Supplementary Fig. 21, the following text **has been added**: “The loading of Pt=N₂=Fe ABA catalyst was 0.08 mg cm^{-2} to avoid overloading.”

Fig. N3 a,b, The ORR polarization plots (a) and the parameters of selectivity (b) of Pt=N₂=Fe ABA catalyst with different loading amount.

5. Question: How did the authors derive the Tafel plots in Fig 2c? The range of potential is inconsistent with the data in Fig 2a. Also, in Fig 2e, for the diffusion-controlled region, you do not use the K-L equation; you use the Levich equation instead. There is no point of plotting the data at various potential positions within the diffusion-controlled region.

Author Reply: Thank you for your careful inspection. We will make a point-by-point response to your questions in the following.

(1) The equation of Tafel slope is $\eta = b \log|j| + a$, which reflects the relationship between the overpotential and the kinetic current density. For the oxygen reduction reaction, the current density (j) is replaced by the kinetic current density (j_k) in the potential range of kinetic-controlled region (from the initial potential to near the half-wave potential) to better reflect the kinetic rate of the reaction. And the j_k is

calculated by the K-L equation: $\frac{1}{J} = \frac{1}{J_L} + \frac{1}{J_K}$. Therefore, based on the polarization curves in Fig. N4a, using the potential (E) and the derived $\log|j_k|$ as the Y and X axes, respectively, the Tafel plots can be obtained. The slope of the Tafel curves reflect the kinetic rate of the oxygen reduction reaction, which can be obtained by linear fitting of the curves. The corrected Tafel plots of the catalysts are shown in Fig. N4b.

Accordingly, Fig. N4b **has been added** in revised manuscript as Fig. 2c, and in line 6, page 9 of the revised manuscript, the following text **has been added**: “The superior ORR kinetics of Pt=N₂=Fe ABA is further confirmed by the smaller Tafel slope (53 mV dec⁻¹), which is significantly smaller than those of commercial Pt/C (84 mV dec⁻¹), Fe AMS (75 mV dec⁻¹) and Pt AMS (90 mV dec⁻¹) (Fig. 2c).”

Fig. N4 a,b, The ORR polarization plots (a) and Tafel plots (b) of Pt=N₂=Fe ABA and the reference catalysts in O₂-saturated 0.1 M KOH.

(2) For the K-L equation:

$$\frac{1}{J} = \frac{1}{J_L} + \frac{1}{J_K} = \frac{1}{B\omega^{1/2}} + \frac{1}{J_K} \quad (1)$$

$$B = 0.62nFC_0D_0^{2/3}V^{-1/6} \quad (2)$$

It can be seen that the Levich equation is the case where $J_k \rightarrow \infty$ in the K-L equation. In the diffusion-controlled region ($J_k \rightarrow \infty$), the limiting diffusion current can reflect the selectivity of the active site to the reaction pathway. As the reviewer stated, we employed the Levich equation in the diffusion-controlled region, fitting the electron transfer number (n) by testing the limiting diffusion current at different rotation speeds. [A.J. Bard, et al. *New York: wiley.* **1980**, 2; *Nat. Mater.* **2011**, 10, 780-786] Based on the polarization curve results at different rotation speeds (Fig.

N5a), we selected a representative potential at 0.70 V vs. RHE to establish the relationship of Levich equation between the limiting diffusion current and the rotation speed, and then obtained the electron transfer number (n) by linear fitting (inset of Fig. N5a). Other three groups data of $(\omega^{-1/2}, J_L^{-1})$ at different potentials in diffusion-controlled region was additionally plotted as the confirmatory results in Fig. N5b, which show an electron transfer number of ~ 3.98 consistent with that at 0.7 V vs. RHE. This result indicates that the Pt=N₂=Fe ABA sample catalyzes the oxygen reduction reaction through a four-electron pathway.

Accordingly, Fig. N5a **has been added** in revised manuscript as Fig. 2e, and Fig. N5b **has been added** in Supplementary Information as Supplementary Fig. 17. In reverse line 7, page 9 of the revised manuscript, the following text **has been added**:
“The ORR pathway was assessed via polarization curve measurements at different rotation rates (Fig. 2e). The electron transfer number (n) was calculated to be approximately 3.98 according to the Levich equation applied in the diffusion-controlled region (inset of Fig. 2e and Supplementary Fig. 17), proving that the Pt=N₂=Fe ABA catalyst favours the 4e⁻ pathway.”

Fig. N5 a,b, Polarization plots at regular rotation rates for Pt=N₂=Fe ABA (a), and the derived Levich plots at 0.70 V (inset in a) and different potentials (b).

6. Question: The ZAB performance, while good, is not exceptional as compared to the state of the art. For instance, in a study by Liao et al (ACS Appl. Mater. Interfaces 2019, 11, 47, 44153–44160), they reported an ultrahigh power density of 479.1 mW cm⁻², more than twice that reported here. Therefore, the question of whether the scaling relation was broken remains unanswered, as one would anticipated a much

enhanced performance.

Author Reply: Thank you for your nice question. Actually, the linear scaling relation that is regulated based on the dual-sites mechanism essentially boosts the catalytic activity of each active site units, which has been reflected in the significantly improved intrinsic activity (turnover frequency, mass activity and selectivity, etc.) compared to the reported ORR catalysts. The mass activity achieved in Pt=N₂=Fe ABA exceeds that of the best platinum-based catalyst reported previously (12.36 A mg_{Pt}⁻¹) [*Science* **2018**, 362, 1276–128] and nearly doubles the outstanding reported mass activity value [*Nat. Catal.* **2022**, 5, 513-523]. However, it is known that the limited metal loading capacity is the main factor that hinders the performance of the single-atom catalysts in fuel cells [*Nat. Commun.* 2021, 12, 5984]. The density of the atomically dual-site centers in the Pt=N₂=Fe ABA catalyst (~ 0.4 at%) is much less than the metal sites density in the metal nanoparticle catalysts, which prevents this dual-sites catalyst from exerting ultrahigh power density in fuel cells. For the metal nanoparticle catalysts (including metal alloy, oxide, etc.), the NCCo-600 catalysts mentioned by the reviewer achieves the optimal power density reported so far, which is more than twice that of ours. However, the loading of the active metal in NCCo-600 is 2.81 at%, which is almost an order of magnitude higher than ours. [*ACS Appl. Mater. Interfaces* **2019**, 11, 44153–44160] If the active metal loading of our single atomic catalysts can be further increased, the power density in fuel cells will be greatly improved, which still remains unsolved.

The highlight of this work is to employ the in-situ synchrotron radiation characterization (XAFS and SR-FTIR) to comprehensive understanding the ORR mechanisms and reveal that O-O radical are promoted for generation and cleavage on well-designed Pt-N-Fe dual-sites. This unique dual-sites mechanism regulates the inherent scaling relation within ORR and endows the catalytic active centers with ultrahigh intrinsic activity. In addition to improving the activity of each metal site, high loading of the active site is required for the single-atom catalysts to achieve the sufficient performance in fuel cells, which is also one of the subjects that we will

strive to overcome in the future. But still, we greatly appreciate your nice question that has actively inspired our in-depth thinking and understanding.

7. Question: The authors need to discuss/mention all figures (in particular those in the SI) in proper order.

Author Reply: Thank you for your kind reminder that is very important to increase the readability of the manuscript. In the revised manuscript, all the Figures and Tables in main text and SI have been mentioned in a logical order.

To Reviewer #2:

General Comment: The authors have made substantial efforts to address the points raised by this reviewer and other reviewers. The revised manuscript appears to be significantly improved compared to the previous version. This reviewer recommend the publication of this paper.

Author Reply: Thank you for your previous constructive criticisms and suggestions, following which the quality of this work has been significantly improved. And we greatly appreciate your approval of our revised manuscript for publication now.

To Reviewer #3:

General Comment: The authors did a thorough re-evaluation of their paper. Especially with the softening of their claims of confinement I believe that they have adequately addressed all of my previous concerns.

Author Reply: Thank you for your previous in-depth and helpful comments, which significantly improve the quality of our manuscript. And we greatly appreciate your approval of our revised manuscript for publication now.

REVIEWERS' COMMENTS

Reviewer #1 (Remarks to the Author):

the revision is good to go